# Accelerated Gradient Descent for Faster Convergence with Minimal Overhead

## Abstract

In this paper, we present CT-AGD (Curvature-Tuned Accelerated Gradient Descent), an optimization method for non-convex optimization problems in deep learning training tasks. CT-AGD is a general boosting procedure that accelerates first-order methods by explicitly capturing the local curvature using finite-difference quotients, and the development of heuristics aimed at mitigating noise and bias introduced by stochastic mini-batch training. CT-AGD has a comparable storage and computational overhead as adaptive gradient methods such as Adam. Our extensive experiments demonstrate that CT-AGD achieves the same level of accuracy as the baseline first-order methods, yet reduces the required training epochs by 33% on average.

## 1 Introduction

We investigate the optimization methods in the training of deep neural networks:

$$\theta^{\star} = \arg\min_{\theta \in \mathbb{R}^d} \mathcal{L}(\theta) = \arg\min_{\theta \in \mathbb{R}^d} \mathbb{E}_{(x,y)\sim\mathcal{P}}\big[\,\ell(f_\theta(x), y)\,\big] \tag{1}$$

where $f_\theta : \mathcal{X} \to \mathcal{Y}$ represent the neural network, $\theta \in \mathbb{R}^d$ are trainable parameters, $\ell(\cdot)$ is the point-wise loss function, and $\mathcal{P}$ is the data distribution. The core of training tasks is essentially the optimization of a large-scale non-convex loss function, requiring substantial computational resources and energy (Strubell et al., 2019; Schwartz et al., 2020; Patterson et al., 2022). Despite significant progress in accelerator hardware (Sze et al., 2020; Jouppi et al., 2017; Hennessy & Patterson, 2019), the optimizers remain critical to reduce end-to-end training time while ensuring stable convergence (Bottou et al., 2018).

Although higher-order, particularly second-order, optimization methods can achieve faster convergence under smoothness and convexity assumptions (Nesterov, 2004; Boyd & Vandenberghe, 2004; Bubeck, 2015), their theoretical advantages are often diminished in deep learning training due to the non-convexity of the loss function landscape, as well as stochastic noises from mini-batch sampling and data augmentation drift (Yao et al., 2021; Bollapragada et al., 2018). First-order methods (SGD and its variants) remain popular in deep learning training due to their robustness and relatively low computational cost (Goyal et al., 2017; Smith et al., 2018).

Adaptive gradient methods using exponential moving averages have shown great success in addressing some key shortcomings of the first-order optimization methods, particularly poor convergence due to the noisy gradient from mini-batching sampling and the sensitivity of learning rates to different scales of widely different curvature. Among them, ADAM (Kingma & Ba, 2015), RM-SProp (Tieleman & Hinton, 2012) and ADADelta (Zeiler, 2012) have been deployed in the successful training of many applications. However, as pointed out in Reddi et al. (2018b); Zaheer et al. (2018a), ADAM-family of adaptive gradient methods may fail to converge to the optimal solution in some relatively simple convex optimization problems when a constant mini-batch size is used. Improvements were also proposed in Zaheer et al. (2018a) to address the convergence issues with better adaptivity of the learning rate. Implicitly, the adaptive gradient methods can be considered as an approximation of the diagonal entries of the full Hessian matrix (or *local curvature*), and use the information to provide better learning rates for different dimensions, versus a global learning rate in the first-order methods.

In this paper, we take a more direct approach by calculating approximations to the diagonal entries of the second-order derivatives (or *quotients*, or *second differences*) during the training steps. Our

proposed method, *Curvature-Tuned Accelerated Gradient Descent* or CT-AGD, accumulates the second differences that are lightweight approximations of the local curvature. Within each epoch, CT-AGD behaves the same as a standard first-order optimizer. In parallel, CT-AGD accumulates per-coordinate quotients, which serve as approximations of local curvatures. Clamp functions are applied to guard the quotients into positive intervals, which suppress oversized steps, and to ensure optimization is stable. At the end of each epoch, CT-AGD performs one additional first-order update, using the quotient accumulation, which can be viewed as the approximation of the diagonal of the Hessian, and provides a direct measure of the local curvature. These direct estimates of the local curvature are also used in the subsequent epoch for faster convergence.

**Contributions** The main contributions of this paper are as follows:

1. We introduce a simple, memory- and compute-efficient optimizer, **CT-AGD**, that can be deployed with existing first-order methods (e.g., SGD, ADAM, etc) with a once-per-epoch curvature estimation. By deploying CT-AGD the optimization method can capture the local curvature while preserving first-order scalability.
2. We provide the theoretical analysis of CT-AGD method by proving its general convergence. We also provide analysis on the trade-off between CT-AGD and the first-order methods, both vanilla and the adaptive gradient variants, as well as the second-order methods.
3. We conduct extensive numerical experiments on the effectiveness and efficiency of CT-AGD. We also provide an intuitive visualization and benchmarking testbed on DL-like objectives. We hope this testbed can be valuable in supporting transparent and reproducible evaluation of future research activities in this direction.

## RELATED WORK

**First-order methods.** Stochastic gradient descent (SGD) remains the cornerstone of large-scale NN training because of its low per-iteration cost and strong anytime performance. Practical accelerations include momentum (Polyak, 1964) and Nesterov's acceleration (Nesterov, 2004), as well as adaptive methods that maintain running statistics of gradients to adjust coordinate-wise step sizes (without modelling curvature explicitly). Representative examples are AdaGrad (Duchi et al., 2011), RMSProp (Tieleman & Hinton, 2012), AdaDelta (Zeiler, 2012), AdamW (Loshchilov & Hutter, 2019), NAdam (Dozat, 2016), Yogi (Zaheer et al., 2018b) and Adam (Kingma & Ba, 2015). Empirically, such methods are competitive across many workloads, but can be sensitive to hyperparameters, and they may converge to an approximate second-order critical point (Wilson et al., 2017; Ruder, 2016; Keskar et al., 2017; Reddi et al., 2018a; Levy, 2016; Jin et al., 2017; Ge et al., 2015). Nonetheless, rigorous nonconvex analyses have been done recently for many of these methods (Reddi et al., 2018b; Ward et al., 2019; Zaheer et al., 2018b). Our work follows this line but uses a *quotient-based* diagonal scaling tied explicitly to curvature proxies gathered during the preceding epoch.

**Second-order, natural-gradient, and quasi-Newton methods.** Quasi-Newton methods approximate inverse curvature from observed changes during optimisation. The canonical example is limited-memory BFGS (L-BFGS), which stores a short history of curvature pairs $(s_i, y_i)$ to build a low-rank inverse–Hessian and applies it with the two-loop recursion without forming matrices (Liu & Nocedal, 1989; Schraudolph et al., 2007; Wang et al., 2017). In deep learning, high dimensionality and mini-batch noise motivate stochastic variants that add damping to control indefiniteness, partition parameters into blocks to reduce memory, and refresh curvature only periodically. These variants can also use Hessian–vector products to apply curvature implicitly, preserving tractability while retaining curvature-informed directions (Pearlmutter, 1994; Byrd et al., 2016; Goldfarb et al., 2020; Anil et al., 2020; Gupta et al., 2018; Yao et al., 2021).

A complementary line of work replaces the Hessian with statistical surrogates. Natural gradient uses the Fisher information as a metric and preconditions gradients by an estimate of $F^{-1}$, which improves conditioning and provides parameterisation invariance (Amari, 1998). Hessian-free and generalised Gauss–Newton methods construct a positive semidefinite curvature matrix such as the GGN or Fisher, and compute search directions with conjugate gradient using Hessian–vector products (Martens, 2010; Schraudolph, 2002). Kronecker-factored approximations such as K-FAC approximate layer-wise Fisher blocks with Kronecker products of small factors, yielding a tractable block-diagonal preconditioner that lowers memory and compute cost and scales to large models (Martens & Grosse, 2015; Grosse & Martens, 2016).

## 2   LOSS OPTIMIZATION IN NEURAL NETWORKS

PROBLEM STATEMENT

Consider the training task of a deep learning network as described in Eqn. (1). Let the training set be $\{(x_i, y_i)\}_{i=1}^{I}$. Consider a depth-$L$ feedforward network with parameters

$$\theta = \{W_\ell, b_\ell\}_{\ell=1}^{L}, \tag{2}$$

where each layer $\ell$ has a weight matrix $W_\ell$ and a bias vector $b_\ell$. In the forward pass step, given an input $x$, the activations are defined recursively:

$$a_0 = x, \qquad h_\ell = W_\ell a_{\ell-1} + b_\ell, \qquad a_\ell = \phi_\ell(h_\ell), \quad \ell = 1, \ldots, L. \tag{3}$$

Here $h_\ell$ are the *pre-activations* (linear combinations of inputs plus bias), and $\phi_\ell(\cdot)$ is a chosen nonlinear activation function. For a per-example loss $\ell(z, y)$ evaluated at the output $z = a_L$, backpropagation computes derivatives backward as follows:

$$\mathcal{D}a_L \;\leftarrow\; \left.\frac{\partial\,\ell(z,y)}{\partial z}\right|_{z=a_L}, \tag{4}$$

and for $\ell = L, \ldots, 1$:

$$g_\ell = \mathcal{D}a_\ell \odot \phi'_\ell(h_\ell), \qquad \nabla W_\ell = g_\ell a_{\ell-1}^\top, \qquad \nabla b_\ell = g_\ell, \qquad \mathcal{D}a_{\ell-1} = W_\ell^\top g_\ell. \tag{5}$$

FIRST- VS SECOND-ORDER OPTIMIZATION METHODS (WITH ADAM/YOGI)

First-order algorithms such as SGD, and the variants with EMA enhancements such as RMSProp, and Adam update parameters using stochastic gradients,

$$\theta_{t+1} = \theta_t - \eta \nabla L(\theta_t), \tag{6}$$

with adaptivity and momentum altering the effective step sizes and directions. These methods are memory efficient and map well to accelerators and mini-batch pipelines, which is why they dominate large-scale training. Their main limitation is sensitivity to ill-conditioning and plateaus in the loss landscape (Ruder, 2016; Yao et al., 2021). By contrast, classical second-order methods like Newton and Levenberg–Marquardt incorporate curvature via (approximations to) the Hessian and can converge rapidly near minima (Hagan & Menhaj, 1994). In deep learning, however, forming or inverting dense curvature is prohibitive, and the stochasticity of mini-batches undermines accurate curvature estimation, which diminishes their practical effectiveness at scale (Ampazis & Perantonis, 2002). Structured approximations (e.g., block diagonal and Kronecker factorizations (Martens & Grosse, 2015)) and stochastic quasi-Newton variants narrow the gap but still face stability and overhead trade-offs in realistic DNN training tasks.

**L-BFGS in Brief**   Limited-memory BFGS (L-BFGS) maintains a short history of curvature pairs $(s_\ell, y_\ell)$ with $s_\ell = \theta_{\ell+1} - \theta_\ell$ and $y_\ell = \nabla L(\theta_{\ell+1}) - \nabla L(\theta_\ell)$, defining an implicit inverse-Hessian $H_t^{-1}$. The two-loop recursion computes the search direction,

$$p_t = -H_t^{-1} \nabla L(\theta_t), \qquad \theta_{t+1} = \theta_t + \alpha_t p_t = \theta_t - \alpha_t H_t^{-1} \nabla L(\theta_t). \tag{7}$$

The two-loop recursion applies $H_t^{-1}$ to a vector in $O(r\,d)$ time using $r$ stored pairs and $d$ parameters (with $r$ typically 5–20 in deep nets). The step length $\alpha_t$ is then chosen separately, usually by a line-search which requires $K - 1$ extra forward–backward passes.

**Adam/Yogi in Brief**   Adam and Yogi maintain exponential moving averages of the gradient and its squared magnitude, with $g_t = \nabla L(\theta_t)$,,

$$m_t = \beta_1 m_{t-1} + (1 - \beta_1)g_t, \tag{8}$$
$$\text{Adam: } v_t = \beta_2 v_{t-1} + (1 - \beta_2)g_t^2, \quad \text{Yogi: } v_t = v_{t-1} - \text{sign}(v_{t-1} - g_t^2)(1 - \beta_2)g_t^2.$$

After bias correction $(\hat{m}_t, \hat{v}_t)$, the search direction and update are

$$p_t = \frac{\hat{m}_t}{\sqrt{\hat{v}_t} + \varepsilon}, \qquad \theta_{t+1} = \theta_t - \eta\,p_t. \tag{9}$$

Per-step memory and compute are $O(d)$. The step length $\eta$ is set externally (often with a schedule), rather than via a line search. AdamW decouples weight decay from the adaptive update; Yogi's variance rule caps unwarranted growth of $v_t$ under noisy gradients.

## 2.1 THE CT-AGD METHOD

The rationale of CT-AGD is to provide more accurate estimates of local curvatures to the optimizers. It can be interpreted as a local-curvature-aware booster of first-order optimizers. Within each mini-batch, it estimates the diagonal of the Hessian, $\widehat{H}$, using finite-difference quotients. At the end of each epoch, it uses this information to perform a more informed update.

**Conventions.** Unless stated otherwise, absolute values, inequalities, divisions, and the clamp $\Pi_{[\lambda_{\min}, \lambda_{\max}]}(\cdot)$ act element-wise; $\oslash$ denotes the element-wise division and $\odot$ denotes the Hadamard product. $K$ denotes the number of epochs with $k$ as the epoch index. $T$ denotes the number of mini-batches in each epoch, with $t$ as the index. Within each mini-batch, the training samples are drawn randomly without replacement with the batch size of $B$.

**Within-Epoch First-Order Steps** In CT-AGD , the optimization within each epoch is almost identical to the classic first-order methods. Therefore, different first-order methods (e.g., SGD and Adam) can be used as the backbone. To simplify annotation, we use vanilla SGD as the example. Starting from $x_{k,0}$, the inner iterations for $t = 0, \ldots, T-1$ are done as follows:

$$\theta_{k,t+1} = \theta_{k,t} - \mu_{k,t} \, g_{k,t}, \qquad \mu_{k,t} = \frac{\eta_1}{\gamma_{k,t}}, \tag{10}$$

with a scalar, iteration-varying *curvature-aware divisor*

$$\gamma_{k,t} = \gamma_k - (\gamma_k - 1)\frac{t}{T}, \qquad t = 0, \ldots, T. \tag{11}$$

With the divisor at the endpoints: $\gamma_{k,0} = \gamma_k$, $\gamma_{k,T} = 1$. (See S.4 for a plot of $\gamma_{k,t}$ over an epoch.)

$\gamma_k$ comes from the curvature estimation of the previous epoch. However, with every step, it loses fidelity as the inner steps move further away from the point of the curvature estimation, hence throughout the epoch:

$$\begin{cases} \gamma_k > 1 : & \mu_{k,t} \text{ starts low and increases linearly to } \eta_1, \\ 0 < \gamma_k < 1 : & \mu_{k,t} \text{ starts high and decreases linearly to } \eta_1, \\ \gamma_k = 1 : & \mu_{k,t} \equiv \eta_1. \end{cases} \tag{12}$$

*Interpretation:* $\gamma_k > 1$ indicates high curvature; $0 < \gamma_k < 1$ indicates low curvature.

**Direct Accumulation of Diagonals of Hessian** Define first differences for $t = 1, \ldots, T-1$,

$$\Delta\theta_{k,t} := \theta_{k,t} - \theta_{k,t-1}, \qquad \Delta g_{k,t} := g_{k,t} - g_{k,t-1}. \tag{13}$$

To ensure the stability, we define a validity mask and the per-coordinate quotients,

$$m_{k,t} := \begin{cases} 1 & \text{if } |\Delta\theta_{k,t}| > \varepsilon \\ 0 & \text{if } |\Delta\theta_{k,t}| \leq \varepsilon \end{cases} \qquad h_{k,t} := \frac{\Delta g_{k,t}}{\Delta\theta_{k,t}} \text{ (element-wise).} \tag{14}$$

We use the following projection operator (or clamp function) to limit the range of computed quotients:

$$\Pi_{[a,b]}(x) = \underset{y \in [a,b]}{\arg\min} |y - x| = \min\big(\max(x,a),\, b\big). \tag{15}$$

The clamped diagonal entries of Hessian are computed as:

$$\widehat{H}_k := \Pi_{[\lambda_{\min}, \lambda_{\max}]}\left(\frac{\sum_{t=1}^{T-1} t \cdot (m_{k,t} \odot h_{k,t})}{(\sum_{t=1}^{T-1} t \cdot m_{k,t}) + \varepsilon}\right) \in \mathbb{R}^d \text{ (element-wise division).} \tag{16}$$

where $0 < \lambda_{\min} \leq \lambda_{\max} < \infty$ and the division is element-wise. The $t$-weights prioritize later inner steps whose finite-difference quotients are closer to the current state. At the same time, the averaging reduces mini-batch noise (that typically corrupts curvature information) to yield a lower-variance (approximately unbiased) diagonal curvature estimate. This way, the diagonal entries provide direct information about local curvature. In CT-AGD they are applied at the end of each epoch as a correction step

$$P_k = \mathbf{1} \oslash \widehat{H}_k = \big(1/H_{k,1}, \ldots, 1/H_{k,d}\big) \in \mathbb{R}^d. \tag{17}$$

*Note on Memory and Compute:* The approximations in Eqn. (16) can be calculated with rolling accumulators for the (weighted) numerator and denominator, as well as the most recent $(\theta_{k,t}, g_{k,t})$ to form $(\Delta\theta_{k,t}, \Delta g_{k,t})$. Their computation only requires $O(d)$ memory per tensor and avoids history buffers.

**One Step Update each Epoch** At the end of each epoch, we have the most accurate estimate of diagonal of Hessian, which provide us the direct information on local curvature. CT-AGD performs one update:

$$\theta_{k+1,0} = \theta_{k,T-1} - \eta_2\, P_k\, \tilde{g}_k, \tag{18}$$

where the gradients $\tilde{g}_k$ are computed as the rolling accumulation by reusing the same rolling accumulators (weighted sum of $g_{k,t}$ and sum of $t$), keeping memory overhead minimal:

$$\tilde{g}_k = \frac{\sum_{t=0}^{T-1} t \cdot g_{k,t}}{\sum_{t=0}^{T-1} t + \varepsilon} \tag{19}$$

Alternatively, we can use the gradients from the last mini-batch:

$$\tilde{g}_k = g_{k,T-1}, \tag{20}$$

The first method reuses the same rolling accumulators (weighted sum of $g_{k,t}$ and sum of $t$), keeping memory overhead minimal; while using the gradients in the last mini-batch requires less computing but higher-variance.

**Scaling Coefficient for the Next Epoch** We calculate the scaling factor for the next epoch as:

$$\gamma_k := Q_\omega(\widehat{H}_k), \tag{21}$$

Here $Q_\omega(\cdot)$ is computed over the entries of the per-tensor diagonal Hessian estimate $\widehat{H}_k$; i.e. we compute the low-tail $\omega$-quantile across its coordinates (elements). Using a low-tail quantile biases $\gamma_k$ toward small curvature directions (small diagonal entries), accelerates the effective learning rate when curvature is low, while remaining robust to outliers. In practice, $\gamma_k$ is computed separately for each tensor/layer, using that tensor's diagonal Hessian estimate, and since $\widehat{H}_k$ is bounded by $\lambda_{mim}$ and $\lambda_{max}$ so is $\gamma_k$; if the quantile is numerically undefined, we set $\gamma_k = 1$.

**Overall Algorithm** The overall algorithm of CT-AGD is summarized below:

---

**Algorithm 1** CT-AGD

---

**Input:** $\theta_0 \in \mathbb{R}^d$, epochs $K$, inner steps $T$, batches $B_t$, steps $\eta_1, \eta_2$, clamp $[\lambda_{\min}, \lambda_{\max}]$, percentile $\omega$, $\varepsilon > 0$, mode $\in \{\texttt{avg}, \texttt{last}\}$, First-order (FO) backbone method.

1: $\gamma \leftarrow 1$
2: **for** $k = 0..K-1$ **do**
3: $\quad$ $S_{\text{num}}, S_{\text{den}} \leftarrow 0$; $\theta \leftarrow \theta_0$
4: $\quad$ **for** $t = 0..T-1$ **do**
5: $\quad\quad$ sample $B_t$; $g_t \leftarrow \nabla f_B(\theta)$
6: $\quad\quad$ $\mu_t \leftarrow \eta_1 / \big(\gamma - (\gamma-1)\frac{t}{T}\big)$ $\quad$ $\theta \leftarrow \text{FO}_{\text{step}}(\theta, g_t, \mu_t)$
7: $\quad\quad$ **if** $t \geq 1$ **then**
8: $\quad\quad\quad$ $m \leftarrow \mathbf{1}_{\{|\theta - \theta_-| > \varepsilon\}}$; $h_t \leftarrow m \odot \frac{g_t - g_-}{\theta - \theta_-}$
9: $\quad\quad\quad$ $S_{\text{num}} \mathrel{+}= t\, h_t$; $S_{\text{den}} \mathrel{+}= t\, m$
10: $\quad\quad$ **end if**
11: $\quad\quad$ $\theta_- \leftarrow \theta$; $g_- \leftarrow g_t$
12: $\quad$ **end for**
13: $\quad$ $\widehat{H} \leftarrow \Pi_{[\lambda_{\min}, \lambda_{\max}]}\big(\frac{S_{\text{num}}}{S_{\text{den}} + \varepsilon}\big)$; $P \leftarrow \mathbf{1} \oslash \widehat{H}$
14: $\quad$ $\gamma \leftarrow Q_\omega(\widehat{H})$ (fallback 1)
15: $\quad$ $\tilde{g} \leftarrow \big(\sum_{t=0}^{T-1} t\, g_t\big) / \big(\sum_{t=0}^{T-1} t + \varepsilon\big)$ if $\texttt{avg}$ else $g_{T-1}$
16: $\quad$ $\theta \leftarrow \theta - \eta_2\, P\, \tilde{g}$
17: **end for**

---

**Updating Scheme when ADAM is used** When the first-order method is ADAM, we replace the first-order step (Eqn. 10) with the bias-corrected ADAM update (Eqn. 9) while leaving the cur-

vature tracking and quantile scaling unchanged. The diagonal curvature estimate $\widehat{H}_k$ (and masking/clamping) is computed exactly as in the SGD case using $\{g_t\}$, and the per-tensor scaling coefficient remains $\gamma_k := Q_\omega(\widehat{H}_k)$ (fallback 1). The epoch-level diagonal estimate is still $P = \mathbf{1} \oslash \widehat{H}_k$, and the outer update uses the stored gradients (not ADAM moments) for $\tilde{g}$:

$$\tilde{g} := \frac{\sum_{t=0}^{T-1} t\, g_t}{\sum_{t=0}^{T-1} t + \varepsilon} \quad \text{if } \texttt{avg}, \quad \text{else} \quad \tilde{g} := g_{T-1}, \qquad \theta \leftarrow \theta - \eta_2\, P\, \tilde{g}.$$

In practice, all quantiles and hessian diagonal estimates are computed per tensor/layer.

**Hyperparameters** A complete list of hyper-parameters and their default values can be find in Supplemental Section S.8.

## 2.2 CONVERGENCE OF CT-AGD

We cast the convergence proof of CT-AGD as outlined in Eqn. (22). Note that we use $x$ as the variable name instead of $\theta$ to avoid confusion.

$$f(x) = \frac{1}{n} \sum_{i=1}^{n} f_i(x), \qquad x \in \mathbb{R}^d, \tag{22}$$

where each $f_i$ is convex and has $L_i$-Lipschitz gradient (i.e., is $L_i$-smooth). Let $L_{\max} := \max_i L_i$. Assume $f$ attains a minimum and let $x^\star \in \arg\min f$ with $f^\star := f(x^\star)$. At iteration $t$, SGD samples $i_t \in \{0, \ldots, T\}$ uniformly at random and performs

$$x_{t+1} = x_t - \eta_1 \nabla f_{i_t}(x_t), \tag{23}$$

where $(\eta_1)_{t \geq 0}$ are step sizes.

We will also use the (at-optimum) gradient noise level

$$\sigma_f^\star := \operatorname{Var}(\nabla f_i(x^\star)) = \mathbb{E}\big[\|\nabla f_i(x^\star)\|^2\big] - \|\mathbb{E}[\nabla f_i(x^\star)]\|^2 = \mathbb{E}\big[\|\nabla f_i(x^\star)\|^2\big], \tag{24}$$

since $\nabla f(x^\star) = \frac{1}{n} \sum_i \nabla f_i(x^\star) = 0$ by optimality (convex and smooth).

CT-AGD introduces effective step sizes $\mu_{k,t} = \eta_1/\gamma_{k,t}$. At the end of each epoch ($k = 1, \ldots, K$) we form a diagonal $P_k = \mathbf{1} \oslash \widehat{H}_k$ from a clamped diagonal secant proxy $\widehat{H}_k \in [\lambda_{min}, \lambda_{max}]^d$ (element-wise), and take one stochastic step with a *fresh* sample:

$$x_{k+1,0} = x_{k,T} - \eta_2 P_k \tilde{g}_k, \qquad \tilde{g}_k := \nabla f_i(x_{k,T}). \tag{25}$$

**Theorem 2.1** (CT-AGD Convergence). *Assume the step sizes satisfy*

$$\eta_1 L_{max} \leq \frac{\lambda_{\min}}{4}\quad {}^{1} \qquad and \qquad \eta_2 \leq \frac{\lambda_{\min}}{4 L_{\max} \lambda_{\max}^2}.$$

*We index epochs by $k = 1, \ldots, K$ and within-epoch steps by $t = 0, \ldots, T$; unrolling via $\ell = (k-1)T + t$ gives a single sequence $\ell = 0, \ldots, N$ with $N = KT$ and define weights as:*

$$a_\ell = \begin{cases} \mu_{k,t}, & \text{if } \ell \text{ is an inner step } (k,t), \\ \lambda_{\min}\eta_2, & \text{if } \ell \text{ is a boundary step } k, \end{cases} \qquad b_\ell = \begin{cases} \mu_{k,t}^2, & \text{if } \ell \text{ is an inner step } (k,t), \\ \lambda_{\max}^2\eta_2^2, & \text{if } \ell \text{ is a boundary step } k. \end{cases}$$

*Then*

$$\mathbb{E}[f(\bar{x}_N) - f^\star] \leq \frac{\|x_0 - x^\star\|^2}{\sum_{\ell=0}^{N-1} a_\ell} + \frac{2\sigma_f^\star \sum_{\ell=0}^{N-1} b_\ell}{\sum_{\ell=0}^{N-1} a_\ell}. \tag{26}$$

*In particular:*

- *If $\sigma_f^\star = 0$ (e.g., interpolation) or if the boundary step uses the full gradient, then $\mathbb{E}[f(\bar{x}_N)] \to f^\star$.*
- *Plain SGD is recovered by taking no boundary step (or $P_k = \mathbf{1}$, $\eta_2 = 0$), yielding the standard bound with $a_\ell = \mu_\ell$, $b_\ell = \mu_\ell^2$ (Garrigos & Gower, 2024).*

The proof can be found in Supplemental Section S.2.

---
1

## 2.3 STORAGE AND COMPUTATIONAL COMPLEXITY

We summarize the storage and dominant per-iteration computational cost below. Note $d$ represents the total number of trainable parameters. We use $s = \Delta\theta$ and $y = \Delta g$ uniformly: $s$ is the parameter change and $y$ the corresponding gradient change. For L-BFGS, let $r$ be the history size (number of stored $(s_i, y_i)$ pairs, typically $r \in [5, 20]$).

Table 1: Per-step total complexity (including f–b passes). $d$ = # parameters; $r$ = L-BFGS history size; $K_t$ = # closure (line-search) evaluations at step $t$.

| Algorithm | Total storage | Elem. ops / step | f–b passes / step | Accounting / notes |
|---|---|---|---|---|
| SGD | $O(d)$ | $O(d)$ | 1 | Parameters only; no optimizer state ($\approx d$). |
| SGD + momentum | $O(2\,d)$ | $O(2\,d)$ | 1 | Parameters + one moment buffer ($\approx 2d$). |
| Adam | $O(3\,d)$ | $O(3\,d)$ | 1 | Parameters + two moment buffers $m$, $v$ ($\approx 3d$). |
| CT-AGD[†] | $O(5\,d)$ | $O(5\,d)$ | 1 | Parameters + four $d$-vectors (prev. $\theta$, prev. $g$, rolling means $h$, $\bar{g}$) ($\approx 5d$). |
| L-BFGS[‡] | $O(r\,d)$ | $O(r\,d)$ | $K_t$ | Parameters + history of $r$ pairs $(s_i, y_i)$; two-loop recursion ($\approx (2r+1)d$). |

[†] Extra *per-epoch* work: one low-tail quantile per tensor of size $d_\ell$; with linear-time selection this is $O\left(\sum_\ell d_\ell\right) = O(d)$; with sorting it is $O\left(\sum_\ell d_\ell \log d_\ell\right) \ll O(d \log d)$ since no global sort over $d$ is performed (both are outside per-step costs).

[‡] With a (possibly inexact) line search evaluating $K_t$ closures at step $t$, total f–b passes per step are $K_t$; with a fixed step or no line search, $K_t = 1$.

## 3 A LIGHTWEIGHT VISUALIZATION TESTBED FOR OPTIMIZERS

We use a simple and self-contained testbed to illustrate the effectiveness of the CT-AGD. The testbed is a quadratic equation, added by a number of Gaussian kernels. To mimic the stochasticity and non-stationarity, at each step, additional random Gaussian noises are added to the quadratic equation itself, as well as the magnitude, mean, and standard deviation of the Gaussian kernel. The testbed mimics three key components of NN training: (i) dynamics of stochastic mini-batch, (ii) non-stationarity across steps, and (iii) the generalization gap between train and test. The testbed, which will be publicly available after the paper is published, can be used to test various gradient-based optimizers. Its details are in Supplemental Section S.3.

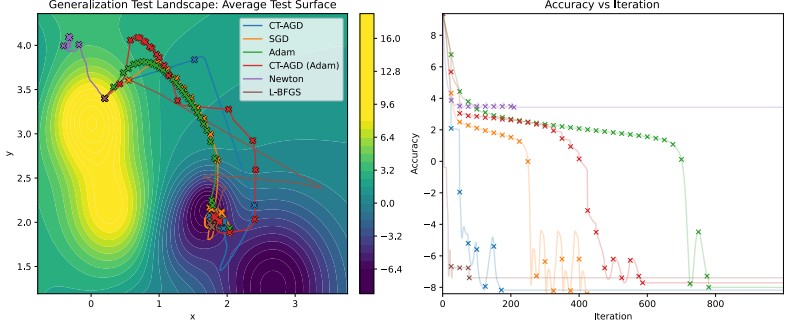

Figure 1: LEFT: Illustration of convergence of CT-AGD, SGD, Adam, Newton and L-BFGS where each step is shown. RIGHT: test accuracy versus iterations. The advange of CT-AGD is clear.

The trajectories in Fig. 1 illustrate the convergence of all methods in one instance. As shown, CT-AGD, SGD, Adam, and L-BFGS descend into the common basin where the minimum is located. The default CT-AGD (with SGD as the backbone, first-order method) requires the least number of steps to achieve so. CT-AGD with ADAM as the backbone is also faster than vanilla ADAM. The pure second-order method, Newton, failed to converge in this case due to its poor tolerance to noise.

More comprehensive results are shown in Tab. 2. We repeated 15 testbed runs, each with a different set of parameters/initial points, and recorded the optimization results as well as the number of steps required to converge. In addition, we evaluate the *generalization gap*, which is the difference between average of test loss and average of training loss.

Observe that the first-order methods tend to achieve better answers (more negative) than L-BFGS, and in fewer steps. Not surprisingly, L-BFGS requires substantially more steps and appears less

| Optimizer | With Random Components | | | Stationary | | |
|---|---|---|---|---|---|---|
| | Final value | # Steps | Gen. gap | Final value | # Steps | Time |
| CT-AGD | $-4.10 \pm 1.29$ | $117.47 \pm 19.70$ | $2.31 \pm 0.91$ | $-6.33 \pm 1.61$ | $144.93 \pm 25.16$ | $0.49 \pm 0.08$ |
| CT-AGD (Adam) | $-4.61 \pm 1.13$ | $137.47 \pm 27.96$ | $1.66 \pm 0.90$ | $-5.75 \pm 2.15$ | $128.07 \pm 49.40$ | $0.43 \pm 0.16$ |
| SGD | $-3.93 \pm 1.10$ | $138.73 \pm 24.54$ | $2.81 \pm 0.55$ | $-6.52 \pm 1.56$ | $150.73 \pm 15.43$ | $0.52 \pm 0.06$ |
| Adam | $-4.31 \pm 1.30$ | $156.53 \pm 24.35$ | $2.08 \pm 0.76$ | $-5.49 \pm 2.33$ | $152.07 \pm 33.91$ | $0.54 \pm 0.12$ |
| L-BFGS | $-3.87 \pm 1.41$ | $278.93 \pm 189.92$ | $2.15 \pm 0.97$ | $-5.19 \pm 2.48$ | $51.07 \pm 0.13$ | $0.33 \pm 0.14$ |

Table 2: Average of 15 testbed runs with different parameters for five method listed. Note that L-BFGS failed to converge in 3 runs, hence the average of 12 is listed. On the right, each run is stationary because no random Gaussian noises are added.

stable due to its poor tolerance to noise (failed to converge in 3 runs). As a control, the results with no Gaussian random components (hence each run becomes stationary) are listed on the right. As expected, L-BFGS takes advantage of the Hessian information and converges much quicker. However, its final value remains less accurate than the first-order methods. CT-AGD with different backbones converges in the fewest number of steps.

# 4 EXPERIMENTAL RESULTS

We compare CT-AGD with SGD and Adam on three datasets: CIFAR-10, CIFAR-100 and Tiny-ImageNet. For each case, we use three architectures: ResNet (He et al., 2016) and Wide ResNet (Zagoruyko & Komodakis, 2016) on all datasets. Additionally, a transformer architecture, DeiT (Touvron et al., 2021), is used for the Tiny-ImageNet dataset. Due to memory and computational constraints, we only run L-BFGS on the CIFAR-10 dataset using ResNet. Each run is repeated at least five times. For each experiment, we report (i) the best test accuracy and (ii) the number of epochs to convergence, which is defined as the number of epochs to achieve within $5\%$ of the final accuracy of that particular run. All methods share identical data pipelines and data augmentation methods. All experiments are conducted with identical hardware and software configurations, which are outlined in Supplemental Section S.7. The hyperparameters used are listed in Supplemental Section S.8. Finally, to demonstrate the method's effectiveness beyond computer vision, we provide additional evaluations on GraphSAGE in Supplemental Section S.6.

The results are summarized in Tab. 3. The accuracy versus epoch of three model-dataset pairs is also shown in Fig. 2. Our general observation is that CT-AGD achieves the same level of accuracy, but requires drastically fewer number of epochs. A few outliers include TinyImageNet with ResNet-20. It appears that Adam is slightly faster, however, the final testing accuracy is inferior ($44.28\%$ versus $46.24\%$ of CT-AGD).

To provide a more reliable comparison, we present a longitudinal comparison in Tab. 4. For each model-dataset pair, we set the cutoff test accuracy as $5\%$ of the *best* accuracy among all optimization methods. The table reports the number of epochs for each optimization method to reach a given threshold. Note that some entries in Tab. 4 are different from the corresponding values in Tab. 3 because of different accuracy thresholds.

| Model | Optimizer | CIFAR-10 | | CIFAR-100 | | Tiny-ImageNet | |
|---|---|---|---|---|---|---|---|
| | | Acc (%) | # Epochs | Acc (%) | # Epochs | Acc (%) | # Epochs |
| ResNet-20 | CT-AGD | $90.05 \pm 0.36$ | $\mathbf{28.20 \pm 2.22}$ | $\mathbf{64.70 \pm 0.32}$ | $\mathbf{48.00 \pm 2.91}$ | $46.24 \pm 0.52$ | $108.50 \pm 7.50$ |
| ResNet-20 | SGD | $\mathbf{90.35 \pm 0.14}$ | $44.00 \pm 3.29$ | $64.10 \pm 0.53$ | $88.60 \pm 7.43$ | $\mathbf{46.88 \pm 0.49}$ | $120.00 \pm 8.91$ |
| ResNet-20 | Adam | $89.20 \pm 0.38$ | $39.20 \pm 3.09$ | $61.86 \pm 0.55$ | $75.20 \pm 2.69$ | $44.28 \pm 0.60$ | $\mathbf{105.60 \pm 7.78}$ |
| ResNet-20 | CT-AGD (Adam) | $88.82 \pm 0.09$ | $31.40 \pm 2.08$ | $61.57 \pm 0.39$ | $72.00 \pm 6.74$ | - | - |
| ResNet-20 | L - BFGS | $82.79 \pm 6.42$ | $90.60 \pm 29.50$ | - | - | - | - |
| WRN-28-4 | CT-AGD | $\mathbf{93.68 \pm 0.15}$ | $23.00 \pm 0.88$ | $73.84 \pm 0.42$ | $\mathbf{37.80 \pm 3.66}$ | $58.69 \pm 0.48$ | $63.25 \pm 5.42$ |
| WRN-28-4 | SGD | $93.69 \pm 0.19$ | $33.80 \pm 4.25$ | $\mathbf{74.12 \pm 0.44}$ | $67.80 \pm 1.62$ | $58.69 \pm 0.50$ | $96.40 \pm 11.77$ |
| WRN-28-4 | Adam | $93.35 \pm 0.09$ | $40.00 \pm 7.02$ | $74.05 \pm 0.20$ | $100.40 \pm 2.26$ | $57.11 \pm 0.29$ | $80.40 \pm 5.99$ |
| WRN-28-4 | CT-AGD (Adam) | $92.36 \pm 0.23$ | $23.00 \pm 1.24$ | $72.09 \pm 0.24$ | $45.20 \pm 5.51$ | - | - |
| DeiT-12 | CT-AGD | - | - | - | - | $31.87 \pm 0.32$ | $\mathbf{37.00 \pm 2.32}$ |
| DeiT-12 | SGD | - | - | - | - | $31.31 \pm 0.35$ | $47.00 \pm 8.14$ |
| DeiT-12 | Adam | - | - | - | - | $\mathbf{32.61 \pm 0.31}$ | $115.40 \pm 14.76$ |

Table 3: Test accuracy (%) and number of epochs to convergence on three datasets with three model architectures. Best results for a model-data combination are in **bold**. Fig. 2 shows the test accuracy versus epochs of shaded cells.

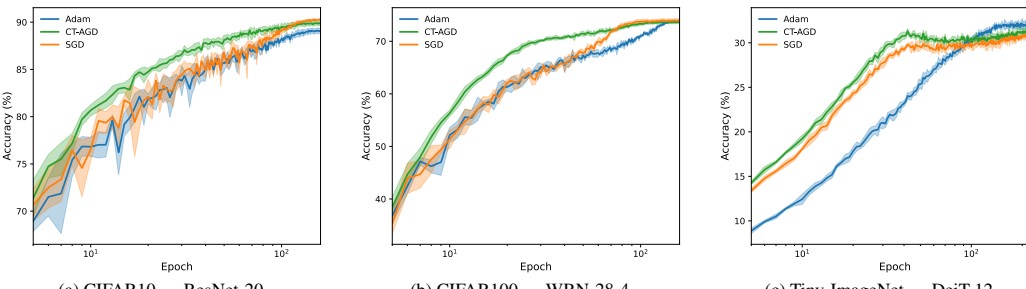

Figure 2: Test accuracy versus epochs of selected model-dataset pairs. See Tab. 3 for more details.

| Model | Dataset | Acc. (%) | CT-AGD | SGD | Adam | CT-AGD (Adam) | Imp. 1 | Imp. 2 |
|---|---|---|---|---|---|---|---|---|
| ResNet-20 | CIFAR-10 | 85.7 | $\mathbf{24.0 \pm 2.9}$ | $32.4 \pm 3.4$ | $39.8 \pm 3.7$ | $34.2 \pm 3.4$ | 25.9% | 14.1% |
| ResNet-20 | CIFAR-100 | 61.2 | $\mathbf{42.2 \pm 1.4}$ | $85.0 \pm 5.0$ | $117.8 \pm 31.2$ | $121.4 \pm 16.8$ | 50.4% | -3.06% |
| ResNet-20 | Tiny-ImageNet | 44.3 | $116.5 \pm 5.5$ | $\mathbf{113.0 \pm 6.4}$ | $145.2 \pm 17.7$ | – | -3.09% | N/A |
| WRN-28-4 | CIFAR-10 | 88.9 | $\mathbf{18.8 \pm 1.6}$ | $22.8 \pm 5.2$ | $28.8 \pm 5.2$ | $23.6 \pm 1.4$ | 17.5% | 18.1% |
| WRN-28-4 | CIFAR-100 | 70.2 | $\mathbf{31.2 \pm 6.0}$ | $62.0 \pm 3.6$ | $83.8 \pm 9.8$ | $57.8 \pm 3.2$ | 49.7% | 31.0% |
| WRN-28-4 | Tiny-ImageNet | 55.5 | $\mathbf{54.5 \pm 9.6}$ | $80.4 \pm 13.8$ | $87.8 \pm 9.7$ | – | 32.2% | N/A |
| DeiT-12 | Tiny-ImageNet | 30.4 | $\mathbf{35.6 \pm 3.7}$ | $91.2 \pm 49.6$ | $96.2 \pm 8.4$ | – | 61.0% | N/A |
| Average | | | | | | | **33.4%** | **15.0%** |

Table 4: Number of epochs for a particular optimization method to reach a common cutoff threshold, which is shown in the third column. The best are shown in **bold**. The first improvement column represents the number of epochs improvement from SGD to CT-AGD with SGD as the backbone. The second improvement column compares CT-AGD with Adam as the backbone with Adam.

Overall we observe $33\%$ cases of fewer number of epochs to achieve the same accuracy. Note that we used the same hyperparameters for all experiments. In two cases when ResNet-20 is used, CT-AGD is slightly slower than the baseline methods. However the difference is within $5\%$. Fine-tuning of hyperparameters can be a solution to move forward. Also note the overfitting in DeiT, which is primarily driven by architectural/dataset. The issue is orthogonal to the optimizer, and should be addressed by other techniques such as data augmentation and regularization. Notice that CT-AGD delivered significant reduction in terms of number of epochs. As shown in Section. 2.3, CT-AGD requires slightly more computation per epoch. Our experiments show that per epoch runtime of CT-AGD is about $9\%$ over its backbone method. However the overhead can be further reduced after code optimization. Overall CT-AGD still provides significant net saving while achieving the same accuracy.

## 4.1 Ablation Studies

We conduct ablation studies on CIFAR-10 (ResNet-20) using the default setup (same as in Tab. 3). We focus on the main heuristic components of the algorithm: (i) clamping of the diagonal curvature proxy, (ii) robustness to noisy curvature quotients, (iii) curvature averaging and decay, and (iv) the low-tail quantile $\omega$.

**Effect of clamping interval.** Tab. 5 reports a sweep of $(\lambda_{\min}, \lambda_{\max})$ spanning three orders of magnitude around the default $(10^{-2}, 10^{2})$.

**Effect of computational noise on the curvature estimate.** Tab. 5 also reports experiments where we inject unbiased Gaussian noise at every mini-batch step. Here what we do is replace Eqn. 14 with

$$h_{k,t} := \frac{\Delta g_{k,t}}{\Delta \theta_{k,t}} + \mathcal{N}(0, \sigma^2) \tag{27}$$

where $\sigma^2$ is varied between experiments.

| Setting | $\lambda_{\min}$ | $\lambda_{\max}$ | Noise $\sigma^2$ | Top-1 (%) | Epochs to conv. |
|---|---|---|---|---|---|
| Interval sweep | $10^{-1}$ | $10^1$ | 0 | $90.13 \pm 0.15$ | $24.60 \pm 2.86$ |
| Interval sweep | $10^{-2}$ | $10^2$ | 0 | $90.01 \pm 0.22$ | $26.20 \pm 4.16$ |
| Interval sweep | $10^{-3}$ | $10^3$ | 0 | $90.23 \pm 0.22$ | $25.80 \pm 1.25$ |
| Noise sweep | – | – | 0 | $90.09 \pm 0.16$ | $23.67 \pm 1.49$ |
| Noise sweep | – | – | 0.01 | $90.07 \pm 0.23$ | $25.80 \pm 2.39$ |
| Noise sweep | – | – | 0.1 | $90.12 \pm 0.14$ | $24.80 \pm 1.04$ |
| SGD baseline | – | – | 0 | $90.35 \pm 0.14$ | $44.00 \pm 3.29$ |

Table 5: Ablations on the clamping interval $(\lambda_{\min}, \lambda_{\max})$ and on injected curvature noise for CT-AGD on CIFAR-10 with ResNet-20. The SGD row is reported for reference.

The results show that both final accuracy and epochs to convergence remain essentially unchanged across a broad range of $(\lambda_{\min}, \lambda_{\max})$, and are also stable under injected noise with variance up to 0.1 at the level of individual curvature quotients, which suggests that CT-AGD is robust.

**Annealing schedule, curvature averaging, and low-tail quantile.** We next ablate the annealing schedule used for propagating curvature information across iterations, the weighting scheme used to average curvature estimates within each epoch, and the low-tail quantile $\omega$ that defines the effective scaling. The first block of Tab. 6 compares: (i) removing annealing entirely by keeping $\mu_{k,t} = \eta_1 / \gamma_{k,t}$ constant through each epoch, (ii) an exponential annealing of the curvature cue with factor $\alpha = \frac{1}{2}$, (iii) the baseline linear annealing with weighted curvature estimation, and (iv) replacing the weighted curvature average by a non-weighted average.

| Setting | Top-1 (%) | Epochs to conv. |
|---|---|---|
| No annealing | $50.96 \pm 3.82$ | $2.20 \pm 0.56$ |
| Exponential annealing ($\alpha = 0.5$) | $90.32 \pm 0.35$ | $27.40 \pm 5.31$ |
| Baseline | $90.01 \pm 0.22$ | $26.20 \pm 4.16$ |
| Non-weighted curvature | $90.05 \pm 0.29$ | $26.80 \pm 1.62$ |
| $\omega = 0.1$ | $90.23 \pm 0.22$ | $25.80 \pm 1.25$ |
| $\omega = 0.2$ | $90.16 \pm 0.26$ | $23.80 \pm 3.21$ |
| $\omega = 0.5$ | $90.18 \pm 0.11$ | $35.80 \pm 3.87$ |
| SGD baseline | $90.35 \pm 0.14$ | $44.00 \pm 3.29$ |

Table 6: Ablations of annealing schedule, curvature averaging, and $\omega$ on CIFAR-10 with ResNet-20.

Removing annealing leads to severe degradation in accuracy and much higher variance, showing that some form of annealing of the inter-epoch curvature cue is essential. This is consistent with our intuition that outdated curvature cues can hinder optimization steps. Both exponential and linear annealing recover high accuracy and similar convergence speed, while the non-weighted curvature average performs almost identically to the weighted version. This indicates that CT-AGD is sensitive to the presence of annealing, but not to a precise functional form or a carefully engineered scheme. The second block of Tab. 6 sweeps the low-tail quantile $\omega$ used to define $\gamma_k$. Across a wide range of values ($0.1 \leq \omega \leq 0.5$), the final accuracy remains essentially unchanged, whereas the number of epochs to convergence mainly reflects how aggressively the method exploits the curvature cue. Smaller values of $\omega$ favor more aggressive updates and faster convergence.

## 5 DISCUSSION

The key insight of CT-AGD is that with careful range limiting and other heuristics, direct calculation of the diagonals of the Hessian can benefit large-scale non-convex optimization problems, which frequently occur in deep neural network training. When compared to adaptive gradient methods (e.g., ADAM), which rely on the moments of the gradients to capture the local curvature, the direct approach deployed in CT-AGD is more responsive to the change of the local curvature, thereby enabling convergence in a smaller number of epochs. Moreover, CT-AGD can be interpreted as a coordinate-wise curvature informed learning rate modulator approach, hence it can be deployed with other first-order optimization methods as its backbone. Our extensive experimental results show that on average CT-AGD achieves in average 33% fewer epochs for the same training task, with up to 61% savings in the best case.

In certain extreme cases (especially with small models on challenging datasets), CT-AGD is slightly slower than the baseline method. ResNet-20 on Tiny-ImageNet is one such example. Nevertheless, CT-AGD still achieves comparable accuracy in this setting, in just 3% more epochs. This little overhead may be reduced with modest hyper-parameter tuning; in practice, adjusting $\lambda_{\min}$, $\lambda_{\max}$, and the quantile threshold $\omega$ can be explored. That said, our ablation studies confirm the method's robustness to these hyperparameter choices, minimizing the need for extensive tuning.

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

## S    SUPPLEMENTAL MATERIAL

### S.1    AUXILIARY LEMMAS

We collect four standard facts used in the proof of Section S.2.

**Lemma S.1** (Unbiasedness of the stochastic gradient)**.** *Under the assumptions above and assuming* $f_i$ *is unbiased, for every* $x$*,*

$$\mathbb{E}[\nabla f_i(x)] = \nabla f(x). \tag{28}$$

**Lemma S.2** (Variance transfer: gradient noise)**.** *Under the assumptions above, for every* $x$*,*

$$\mathbb{E}\big[\|\nabla f_i(x)\|^2\big] \;\leq\; 4L_{\max}\big(f(x) - f^\star\big) \;+\; 2\,\sigma_f^\star. \tag{29}$$

**Lemma S.3** (Convexity pairing)**.** *Under the assumptions above, for every* $x$*,*

$$\langle \nabla f(x),\, x - x^\star \rangle \geq f(x) - f^\star. \tag{30}$$

**Lemma S.4** (Spectral sandwich)**.** *If* $\widehat{H}_k \in [\lambda_{\min}, \lambda_{\max}]$ *elementwise and* $P_k = \mathrm{diag}(1/\widehat{H}_k)$*, then*

$$\lambda_{\min}I \preceq P_k \preceq \lambda_{\max}I, \tag{31}$$

*so* $\langle P_k u, v \rangle \geq \lambda_{\min}\langle u, v \rangle$ *and* $\|P_k u\| \leq \lambda_{\max}\|u\|$ *for all* $u, v$.

### S.2    CT-AGD CONVERGENCE PROOF

*Proof.* of Theorem 2.1.

**Inner (SGD) step.**    Fix $k, t$ and condition on $x_{k,t}$. Let $g_{k,t} = \nabla f_{i_{k,t}}(x_{k,t})$. Expanding the square and taking conditional expectation,

$$
\begin{aligned}
\mathbb{E}\big[\|x_{k,t+1} - x^\star\|^2 \,\big|\, x_{k,t}\big] &= \|x_{k,t} - x^\star\|^2 - 2\mu_{k,t}\,\langle \mathbb{E}[g_{k,t} \,|\, x_{k,t}],\, x_{k,t} - x^\star\rangle + \mu_{k,t}^2\,\mathbb{E}\big[\|g_{k,t}\|^2 \,|\, x_{k,t}\big] \\
&\leq \|x_{k,t} - x^\star\|^2 - 2\mu_{k,t}\,(f(x_{k,t}) - f^\star) + \mu_{k,t}^2\big(4L_{\max}(f(x_{k,t}) - f^\star) + 2\sigma_f^\star\big) \\
&= \|x_{k,t} - x^\star\|^2 + 2\mu_{k,t}\big(2L_{\max}\mu_{k,t} - 1\big)(f(x_{k,t}) - f^\star) + 2\mu_{k,t}^2\,\sigma_f^\star,
\end{aligned}
$$

where we used unbiasedness S.1, convexity pairing S.3, and the variance-transfer bound S.2. Because $\eta_1 L_{\max} \leq \lambda_{\min}/4$, we get $\mu_{k,t}L_{\max} \leq \frac{1}{4}$, and thus $2\mu_{k,t}(2L_{\max}\mu_{k,t} - 1) \leq -\mu_{k,t}$, hence

$$\mathbb{E}\big[\|x_{k,t+1} - x^\star\|^2 \,\big|\, x_{k,t}\big] \leq \|x_{k,t} - x^\star\|^2 - \mu_{k,t}\,(f(x_{k,t}) - f^\star) + 2\mu_{k,t}^2\,\sigma_f^\star. \tag{32}$$

**Boundary step.**    Fix $k$ and condition on $x_{k,T}$ and $P_k$. Use a fresh sample $\tilde{g}_k$ so that $\mathbb{E}[\tilde{g}_k \,|\, x_{k,T}, P_k] = \nabla f(x_{k,T})$. Expand and take conditional expectation:

$$\mathbb{E}\big[\|x_{k+1,0} - x^\star\|^2 \,\big|\, x_{k,T}, P_k\big] = \|x_{k,T} - x^\star\|^2 - 2\eta_2\langle P_k\nabla f(x_{k,T}),\, x_{k,T} - x^\star\rangle + \eta_2^2\,\mathbb{E}\big[\|P_k\tilde{g}_k\|^2 \,|\, x_{k,T}, P_k\big].$$

By the spectral sandwich S.4 and convexity pairing S.3, $\langle P_k\nabla f(x),\, x - x^\star\rangle \geq \lambda_{\min}(f(x) - f^\star)$. Moreover, $\|P_k\tilde{g}_k\|^2 \leq \lambda_{\max}^2\|\tilde{g}_k\|^2$ and the variance-transfer bound S.2 yields

$$\mathbb{E}\big[\|P_k\tilde{g}_k\|^2 \,|\, x_{k,T}, P_k\big] \leq \lambda_{\max}^2\big(4L_{\max}(f(x_{k,T}) - f^\star) + 2\sigma_f^\star\big).$$

Hence

$$\mathbb{E}\big[\|x_{k+1,0} - x^\star\|^2 \,\big|\, x_{k,T}, P_k\big] \leq \|x_{k,T} - x^\star\|^2 + 2\eta_2\Big(2L_{\max}\lambda_{\max}^2\eta_2 - \lambda_{\min}\Big)(f(x_{k,T}) - f^\star) + 2\lambda_{\max}^2\eta_2^2\,\sigma_f^\star.$$

Because $\eta_2 \leq \lambda_{\min}/(4L_{\max}\lambda_{\max}^2)$, then $2\eta_2(2L_{\max}\lambda_{\max}^2\eta_2 - \lambda_{\min}) \leq -\lambda_{\min}\eta_2$, and thus

$$\mathbb{E}\big[\|x_{k+1,0} - x^\star\|^2 \,\big|\, x_{k,T}, P_k\big] \leq \|x_{k,T} - x^\star\|^2 - \lambda_{\min}\eta_2\,(f(x_{k,T}) - f^\star) + 2\lambda_{\max}^2\eta_2^2\,\sigma_f^\star. \tag{33}$$

**Unification and telescoping.** Index all updates by $\ell$ and write a generic one-step inequality

$$\mathbb{E}\big[\|x_{\ell+1} - x^\star\|^2 \,\big|\, x_\ell\big] \leq \|x_\ell - x^\star\|^2 - a_\ell \left(f(x_\ell) - f^\star\right) + 2b_\ell \, \sigma_f^\star,$$

where $(a_\ell, b_\ell)$ are given in the statement (by Eqn. 32 for inner steps and Eqn. 33 for boundary steps). Summing from $\ell = 0$ to $N-1$ and taking total expectation yields

$$\sum_{\ell=0}^{N-1} \mathbb{E}\big[\|x_{\ell+1} - x^\star\|^2\big] - \sum_{\ell=0}^{N-1} \mathbb{E}\big[\|x_\ell - x^\star\|^2\big] \leq -\sum_{\ell=0}^{N-1} a_\ell \, \mathbb{E}[f(x_\ell) - f^\star] \;+\; 2\,\sigma_f^\star \sum_{\ell=0}^{N-1} b_\ell$$

$$\mathbb{E}\big[\|x_N - x^\star\|^2\big] - \|x_0 - x^\star\|^2 \leq -\sum_{\ell=0}^{N-1} a_\ell \, \mathbb{E}[f(x_\ell) - f^\star] \;+\; 2\,\sigma_f^\star \sum_{\ell=0}^{N-1} b_\ell.$$

Rearranging gives

$$\sum_{\ell=0}^{N-1} a_\ell \, \mathbb{E}[f(x_\ell) - f^\star] \;\leq\; \|x_0 - x^\star\|^2 \;-\; \mathbb{E}\big[\|x_N - x^\star\|^2\big] \;+\; 2\,\sigma_f^\star \sum_{\ell=0}^{N-1} b_\ell. \tag{34}$$

Since $\mathbb{E}[\|x_N - x^\star\|^2] \geq 0$, we can drop it.

Then dividing both sides by the positive total weight $S_N := \sum_{\ell=0}^{N-1} a_\ell$ gives us:

$$\frac{1}{S_N} \sum_{\ell=0}^{N-1} a_\ell \, \mathbb{E}[f(x_\ell) - f^\star] \;\leq\; \frac{\|x_0 - x^\star\|^2}{S_N} \;+\; \frac{2\,\sigma_f^\star \sum_{\ell=0}^{N-1} b_\ell}{S_N}. \tag{35}$$

**Jensen's inequality.** If $f : \mathbb{R}^d \to \mathbb{R}$ is convex and $w_0, \ldots, w_{N-1} \geq 0$ with $\sum_\ell w_\ell = 1$, then for any $x_0, \ldots, x_{N-1} \in \mathbb{R}^d$,

$$f\left(\sum_{\ell=0}^{N-1} w_\ell x_\ell\right) \;\leq\; \sum_{\ell=0}^{N-1} w_\ell \, f(x_\ell). \tag{36}$$

Apply Eqn.( 36) with deterministic weights $w_\ell := a_\ell/S_N$ and define the weighted average iterate

$$\bar{x}_N \;:=\; \sum_{\ell=0}^{N-1} w_\ell x_\ell \;=\; \frac{1}{S_N} \sum_{\ell=0}^{N-1} a_\ell x_\ell.$$

And we get

$$\mathbb{E}[f(\bar{x}_N) - f^\star] \;\leq\; \frac{\|x_0 - x^\star\|^2}{\sum_{\ell=0}^{N-1} a_\ell} \;+\; \frac{2\,\sigma_f^\star \sum_{\ell=0}^{N-1} b_\ell}{\sum_{\ell=0}^{N-1} a_\ell}. \tag{37}$$

$\square$

## S.3   Dynamic "Generalization" Landscape

**Snapshots and drift.** The landscape is defined by a sequence of *train* snapshots and a shorter sequence of *test* snapshots, built sequentially to emulate nonstationarity. Each snapshot is a sum of a convex quadratic baseline and a set of drifting Gaussian "lumps" with random signs (to create both attractive and repulsive features). Let $\theta \in \mathbb{R}^2$, let $c_0 \in \mathbb{R}^2$ be the target center, $q > 0$ the quadratic factor, and for $j = 1, \ldots, M$ let $(c_j, a_j, s_j, \sigma_j)$ denote center, amplitude, scale, and sign ($\sigma_j \in \{-1, +1\}$). The value of a snapshot $\mathcal{S}$ at $\theta$ is

$$V(\theta; \mathcal{S}) \;=\; q \, \|\theta - c_0\|_2^2 \;+\; \sum_{j=1}^{M} \sigma_j \, a_j \exp\!\left(-\frac{\|\theta - c_j\|_2^2}{2s_j^2}\right). \tag{38}$$

Snapshots evolve by small stochastic drifts:

$$c_j^{(t+1)} = c_j^{(t)} + \varepsilon_{c,j}^{(t)}, \qquad\qquad \varepsilon_{c,j}^{(t)} \sim \mathcal{N}(0, \sigma_c^2 I_2),$$

$$a_j^{(t+1)} = a_j^{(t)} \, [1 + \varepsilon_{a,j}^{(t)}], \qquad\qquad \varepsilon_{a,j}^{(t)} \sim \mathcal{N}(0, \sigma_a^2),$$

$$s_j^{(t+1)} = s_j^{(t)} \, [1 + \varepsilon_{s,j}^{(t)}], \qquad\qquad \varepsilon_{s,j}^{(t)} \sim \mathcal{N}(0, \sigma_s^2).$$

Calling the landscape once advances the train snapshot index (cyclically), so an optimiser observes a *changing* objective during a trajectory, akin to iterating over mini-batches and data augmentations.

**Train, test, and the gap.** Let $\{\mathcal{S}_t^{\text{train}}\}_{t=1}^{T_{\text{train}}}$ and $\{\mathcal{S}_u^{\text{test}}\}_{u=1}^{T_{\text{test}}}$ be the built sequences. For any point $\theta$, we define

$$L_{\text{avg}}(\theta) := \frac{1}{T_{\text{train}}} \sum_{t=1}^{T_{\text{train}}} V\left(\theta; \mathcal{S}_t^{\text{train}}\right) \quad E_{\text{avg}}(\theta) := \frac{1}{T_{\text{test}}} \sum_{u=1}^{T_{\text{test}}} V\left(\theta; \mathcal{S}_u^{\text{test}}\right) \tag{39}$$
$$\text{Gap}(\theta) := E_{\text{avg}}(\theta) - L_{\text{avg}}(\theta)$$

which play the role of train loss, test loss, and generalisation gap. In visualisations, we may draw either the instantaneous train surface $V(\cdot; \mathcal{S}_t^{\text{train}})$ or the averaged test surface $E_{\text{avg}}(\cdot)$ as the background to contrast optimisation progress and generalisation.

To visualise the animation of how the landscape works and how different optimisers behave (results from Figure 1), follow the link:
*https://osf.io/63zem?view_only=b0f43a3664f44dba98cad7a1c4c33cc2*

### S.4 EVOLUTION OF THE CURVATURE-AWARE DIVISOR

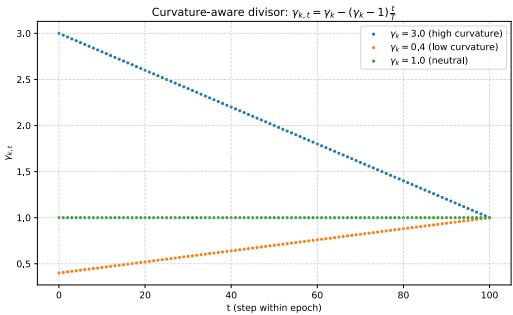

Figure 3: Evolution of the curvature-aware divisor $\gamma_{k,t} = \gamma_k - (\gamma_k - 1)\frac{t}{T}$ over one epoch (of 100 internal steps) for three initial values: $\gamma_k > 1$, $0 < \gamma_k < 1$, and $\gamma_k = 1$. The schedule is linear from $\gamma_k$ at $t = 0$ to 1 at $t = T$.

### S.5 ADITIONAL RESULTS

| Model | Optimizer | CIFAR-10 | | CIFAR-100 | | Tiny-ImageNet | |
|---|---|---|---|---|---|---|---|
| | | Acc (%) | Time (s) | Acc (%) | Time (s) | Acc (%) | Time (s) |
| ResNet-20 | CT-AGD | 90.05 ± 0.36 | **255.13 ± 30.17** | **64.70 ± 0.32** | **463.03 ± 15.86** | 46.24 ± 0.52 | 4343.79 ± 207.09 |
| ResNet-20 | SGD | **90.35 ± 0.14** | 316.81 ± 28.85 | 64.10 ± 0.53 | 857.62 ± 54.83 | **46.88 ± 0.49** | **3660.27 ± 202.14** |
| ResNet-20 | Adam | 89.20 ± 0.38 | 388.12 ± 37.54 | 61.86 ± 0.55 | 1179.86 ± 306.30 | 44.28 ± 0.60 | 4771.47 ± 660.88 |
| ResNet-20 | CT-AGD (Adam) | 88.82 ± 0.09 | 374.33 ± 36.40 | 61.57 ± 0.39 | 1335.25 ± 190.49 | - | - |
| ResNet-20 | L-BFGS | 82.79 ± 6.42 | 16295.83 ± 3140.84 | - | - | - | - |
| Wide ResNet | CT-AGD | 93.68 ± 0.15 | **388.66 ± 33.75** | 73.84 ± 0.42 | **666.94 ± 132.49** | **58.69 ± 0.48** | 9382.40 ± 2322.14 |
| Wide ResNet | SGD | **93.69 ± 0.19** | 416.61 ± 95.57 | **74.12 ± 0.44** | 1140.35 ± 66.81 | **58.69 ± 0.50** | 11015.84 ± 1881.44 |
| Wide ResNet | Adam | 93.35 ± 0.09 | 525.29 ± 93.24 | 74.05 ± 0.20 | 1542.87 ± 185.55 | 57.11 ± 0.29 | 12000.42 ± 1317.28 |
| Wide ResNet | CT-AGD (Adam) | 92.36 ± 0.23 | 524.07 ± 31.57 | 72.09 ± 0.24 | 1283.09 ± 72.73 | - | - |
| DeiT | CT-AGD | - | - | - | - | 31.87 ± 0.32 | **3785.37 ± 415.78** |
| DeiT | SGD | - | - | - | - | 31.31 ± 0.35 | 8598.44 ± 4677.69 |
| DeiT | Adam | - | - | - | - | **32.61 ± 0.31** | 9132.21 ± 799.30 |

Table 7: Test accuracy (%) and *Time to convergence*. Accuracies are the same as in Tab. 3, but convergence is defined as the first epoch at which a run reaches 5% of the *best* max test accuracy among optimizers for the same model–dataset pair. Best results for a model-data combination are in **bold**. Fig. 4 shows the test accuracy versus epochs of shaded cells.

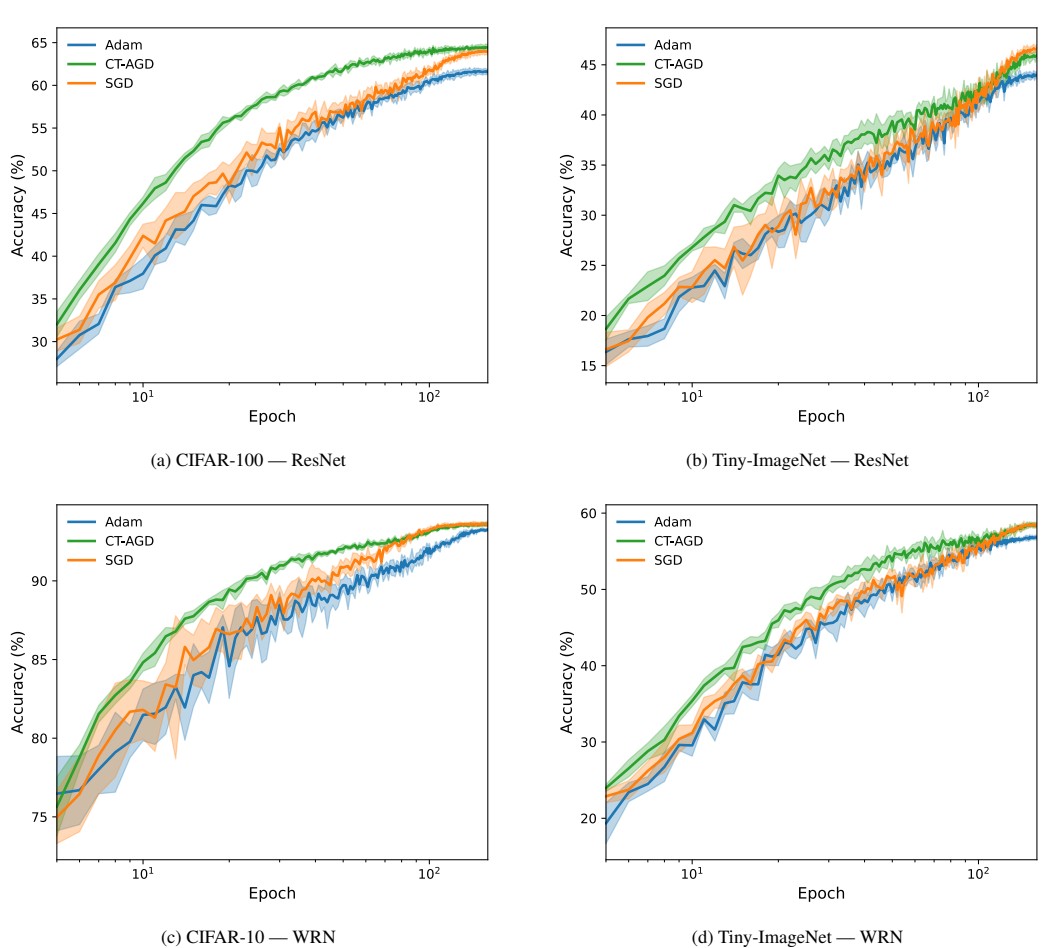

Figure 4: **Additional accuracy trajectories.** Complements Fig. 2 by showing the non-diagonal tasks.

## S.6 ADDITIONAL EXPERIMENTS

We expanded our evaluation by testing CT-AGD on large-scale node classification on the Reddit benchmark. The Reddit graph contains $\sim$ 232k nodes and $\sim$ 11.6M edges with 602-dimensional input features and 41 target classes. We train a 2-layer GraphSAGE over 60 epochs. All optimizers share the same architecture and training protocol. Tab. 8 reports the final training accuracy and the number of epochs to convergence, averaged over 5 runs.

| Method | Train acc. (%) | Epochs to conv. |
|---|---|---|
| Adam | $97.60 \pm 0.00$ | $23.4 \pm 0.5$ |
| CT-AGD (Adam) | $97.60 \pm 0.00$ | $23.2 \pm 0.4$ |
| CT-AGD (SGD) | $93.30 \pm 0.00$ | $18.8 \pm 0.8$ |
| SGD | $92.80 \pm 0.00$ | $27.8 \pm 0.4$ |

Table 8: Reddit node classification with a 2-layer GraphSAGE encoder. CT-AGD (Adam) matches the strong training accuracy of Adam with essentially identical convergence, while CT-AGD (SGD) improves over plain SGD in both training accuracy and the number of epochs required to converge.

## S.7 HARDWARE AND SOFTWARE

Hardware and software configurations for the experimental results.

| Hardware | |
|---|---|
| CPU | Intel Xeon Silver 4214R @ 2.40 GHz |
| vCPU topology | 8 vCPUs; 2 threads/core; NUMA nodes: 2 |
| Memory | 64 GiB RAM (swap: 0 B); HugePages: 0 |
| GPU | NVIDIA Tesla V100S-PCIe-32GB ($1\times$; 32 GB HBM2) |
| Driver | NVIDIA 555.42.02 |
| **Software** | |
| OS / Kernel | Ubuntu 20.04.4 LTS (Focal), Linux 5.4.0-120-generic |
| Python | 3.10.18 |
| PyTorch | 2.7.1 +cu126 |
| Torchvision | 0.22.1 |
| CUDA | 12.6 |

## S.8 HYPER-PARAMETERS

CT-AGD hyper-parameters and values used

- Secondary learning rate: $\eta_2 = 0.5$.
- Clamp bounds: $\lambda_{\min} = 10^{-2}$, $\lambda_{\max} = 10^2$.
- Percentile statistic: $\omega = 0.1$.
- Numerical stabilizer: $\varepsilon = 10^{-3}$.
- Gradient update mode: `avg`.

CT-AGD inherits the hyperparameters of its backbone optimizer: CT-AGD (Adam) uses Adam's defaults; CT-AGD uses SGD's defaults. This means, they use the same values as the standalone optimizer.

| Method | Primary learning rate $\eta_1$ and common defaults |
|---|---|
| Adam and CT-AGD (Adam) | $\eta_1 = 10^{-3}$; $(\beta_1, \beta_2) = (0.9, 0.999)$ |
| SGD and CT-AGD | $\eta_1 = 10^{-2}$; momentum $\mu = 0.85$ (no Nesterov) |
| L-BFGS | initial step size $\eta_1 = 1.0$; history size $r = 10$; max inner iterations $K = 20$ |

All methods use weight decay $\lambda = 5 \times 10^{-4}$.

## S.9 USAGE OF LLM

We have used LLM (ChatGPT) for the following tasks:

1. Search the web to find papers on Adam, Yogi, L-BFGS, RMSProp and other related topics.
2. Polishing the writing of the manuscripts by revising a sentence or part of a sentence to better convey the authors' intention. We also used LLM to suggest better words to describe certain concepts, or different verbs to better express certain concepts.

