# OpenReview forum: "Accelerated Gradient Descent for Faster Convergence with Minimal Overhead"
_ICLR.cc/2026/Conference — Submitted to ICLR 2026_

### Official Review · Reviewer_DdLg · 2025-10-18

**Soundness:** 2
**Presentation:** 2
**Contribution:** 1
**Rating:** 0
**Confidence:** 4

**Summary:**

The paper introduces **CT-AGD (Curvature-Tuned Accelerated Gradient Descent)**, a boost technique for first-order optimizers (e.g., SGD, Adam). During each epoch, it runs standard first-order updates while accumulating per-coordinate finite-difference quotients as diagonal curvature proxies; at the epoch boundary it applies one extra preconditioned step using the clipped/quantile-robust diagonal. Experiments on CIFAR-10/100 and Tiny-ImageNet with ResNet, Wide-ResNet, and DeiT report comparable accuracy with ≈33% fewer epochs on average.

**Strengths:**

The approach is architecturally simple and integrates as a once-per-epoch preconditioning step on top of standard first-order methods (e.g., SGD, Adam). This makes it easy to extend in existing training pipelines without major code or infrastructure changes.

**Weaknesses:**

## Major

1. This paper reads like a recycled submission from several years ago. The reviewed and compared works are largely outdated despite rapid progress in optimizers. For an ICLR 2026 submission, the related work should at least minimally cover recent advances (2023–2025). As concrete evidence: there are zero 2025 citations, only one 2024 citation of a handbook [1], no 2023 citations, and just one 2022 citation [2], and that 2022 citation is a background article on carbon cost rather than an optimization paper. In the Related Work section, all cited papers are 2021 or earlier, which signals a significant gap in coverage of the modern optimizer landscape. Please at least update the discussion to include contemporary adaptive, curvature-aware, and large-scale training optimizers for submission.

2. Section 2’s exposition on neural network structures and backpropagation feels overly general and not directly relevant to the proposed method. Consider trimming or removing this material to keep the problem statement focused.

3. The theoretical setup is oversimplified. Convergence is proved only when the objective is an average of convex functions. For relevance to deep learning, a non-convex analysis under standard assumptions (e.g., $L$-smooth objectives) is expected, as is common in optimizer literature.

4. The convergence results offer limited insight. The analysis assumes a diagonal preconditioner with entries clamped to $[\lambda_{\min}, \lambda_{\max}]$ but does not justify why or when this diagonal meaningfully approximates the Hessian structure. Without such justification, the theory does not explain why the optimizer should be useful in practice.

5. The memory footprint appears large (≈ $5d$ states). This makes the method unlikely to be practical for large-scale neural network training, where memory pressure is already severe.

6. The experiments are mostly at toy scale (e.g., CIFAR-10 with ResNet-20), which limits the external validity of the claims. It remains unclear whether the method is useful in realistic large-model scenarios.

7. CT-AGD is not consistently competitive. In several cases, it is surpassed by vanilla SGD (see Figure 2 and Table 4), which is generally regarded as a weak baseline in modern deep learning optimization.

8. The method introduces numerous hyperparameters, $\eta_1$, $\eta_2$, $\lambda_{\min}$, $\lambda_{\max}$, percentile $\omega$, on top of Adam’s own settings, but there is no systematic guidance or ablation on how to choose them or how sensitive performance is to these choices.

## Minor

- Lines 139–140: the statement “Structured approximations (e.g., block diagonal and Kronecker factorizations) and stochastic quasi-Newton variants narrow the gap...” needs citations; e.g., KFAC [3].
- Figure 1 (right): the title says “accuracy vs. epoch,” but the x-axis is iterations. Please correct the label or the title to avoid confusion.
- The term “accelerated” is typically associated with Nesterov’s accelerated gradient; since the proposed method is unrelated, please explicitly clarify that there is no connection to Nesterov-style acceleration.

## References

[1] Guillermo Garrigos, Gauthier Gidel, and Aymeric Dieuleveut. *Handbook of convergence theorems for (stochastic) gradient methods.* arXiv:2301.11235, 2024.
[2] David A. Patterson, Joseph E. Gonzalez, Urs Hölzle, Quoc V. Le, Chen Liang, Lluis-Miquel Munguia, Daniel Rothchild, David R. So, Maud Texier, and Jeff Dean. *The carbon footprint of machine learning training will plateau, then shrink.* *Computer*, 55(7):18–28, 2022.
[3] Martens, James, and Roger Grosse. *Optimizing neural networks with Kronecker-factored approximate curvature.* ICML, 2015.

**Questions:**

1. Why can Equation 16 be regarded as a Hessian approximation? Please provide a minimal but clear explanation or derivation that links the quotient-based estimate to a diagonal Hessian model, including assumptions under which this interpretation holds.

2. The scaling coefficient for the next epoch appears ad hoc. How does this heuristic compare to Levenberg–Marquardt style adaptive damping in terms of stability, convergence speed, and sensitivity to hyperparameters?

---

> ### Author Response · Authors · 2025-11-20
>
> We thank the reviewer for the careful and detailed feedback.
>
> **Q1**: The approximation is justified by the secant condition from quasi-Newton theory. This is the standard approach and has been validated extensively in optimization literature:
> https://pages.cs.wisc.edu/~yudongchen/cs726_sp23/Lecture_21_quasi_Newton.pdf
>
> **Q2**: It is interesting reviewer has brought up LM. As an optimization method, LM is a second-order method, which generally has better convergence than quasi-Newton's methods, including L-BFGS. Because it requires computation of the Jacobian matrix, it is computationally more expensive than quasi-Newton's method, let alone the gradient descent methods.  More critically, LM is tailored for least-square loss function for nonlinear problems. It is not suitable in general DL settings, which often require other loss functions such as cross-entry, and KL divergence.
>
> **W1**: We don't quite understand why reviewer DdLg brought up this issue. We don't believe the publication years of the references are relavent to the quality of a manuscript. For the record, this mansucript has not been submitted to any other venues.
>
> **W2**: We included the background information to accomodate the broad audience of ICLR. We have made some revisions based on the reviewers' comments.
>
> **W3** and **Q4**: Our theoretical analysis was meant solely to show that the existing guarantees were maintained for convex settings. No additional claims were made or meant for general problems. We have revised the manuscript to insure no such claims exist. There is additional overhead compaired to Adam but it is within the same order of magnitude, which is why we termed it minimal when compared to second order methods such as L-BFGS.
>
> **W5**: We believe our method has modest memory footprint, since it's has linear complexity with respet to the number of parameters. As shown in Table 1, CT-AGD is just slightly more expensive than ADAM on both memory storage and computational. As a reference, the LM method, has the memory complexity of $O(d^2)$.
>
> **W6**: We included CIFAR10 because this is the only example L-BFGS can run, which we would like to include to give the readers a comprehensive view of the performance. We do include a large dataset, tinyImageNet, which has 120,000 datapoints. We have included more datasets and are in the process off running more, to be included in the manuscript.
>
> **W7**: We disagree the statement that CT-AGD is not competitive. Given the stochastical nature of DL training, it is difficult to find one particular optimization method is always suprior. As can be observed in our exeperiments, there is no guarantee that more advanced and slightly more expensive methods such as ADAM always outperform SGD. We faithfully reported our experimental results, so that the readers can have a comprehensive view. We also want to point out that in a few cases CT-AGD is the not the best performaning method, the gap is small, often within the reproducibility margin.
>
> **W8**: We agree that introducing new hyperparameters without presenting ablation studies can raise realistic concerns about the applicability of the method. We have updated the manuscript to include the ablations previously mentioned which provide better confidence in the robustness of the method.
>
> **Minor**: We will revise the manuscript to address the style, notation, and presentation issues raised in the minor comments.

---

### Official Review · Reviewer_FzcT · 2025-10-25

**Soundness:** 3
**Presentation:** 3
**Contribution:** 3
**Rating:** 4
**Confidence:** 4

**Summary:**

This paper introduces CT-AGD, a novel procedure to "boost" first-order optimizers like SGD and Adam for non-convex deep learning tasks. The method's core mechanism is the direct estimation of a diagonal curvature (Hessian) proxy using finite-difference quotients (\Delta g / \Delta \theta) accumulated during each epoch. This estimate is then ingeniously used in two ways: 1) as a pre-conditioner for a single, additional update step at the end of the epoch, and 2) to compute a scalar learning rate modulator, \gamma_k, that scales the intra-epoch steps of the next epoch.

**Strengths:**

The authors present extensive experiments showing that CT-AGD achieves a significant average reduction of 33% in the number of epochs required to reach a target accuracy, while achieving a final accuracy comparable to the baseline optimizers.

**Weaknesses:**

Despite the promising empirical results and the cleverness of the algorithm's design, the paper in its current form suffers from several fundamental weaknesses:

1. The theoretical analysis is critically disconnected from the method's application, providing convex guarantees for a non-convex problem. The algorithmic design is a complex amalgamation of non-trivial heuristics (clamping, weighting, annealing, quantiles) presented without any ablation studies to justify their inclusion or necessity. Finally, the paper's claims of "minimal overhead" are questionable, as the method introduces significant optimizer state memory overhead compared to Adam and a non-trivial number of new, sensitive hyperparameters.

2. The novelty of CT-AGD lies in its specific, dual-use of a direct, secant-based diagonal curvature estimate. However, the stability of this estimate is the central challenge, and the paper attempts to solve it by introducing a cascade of heuristics. The lack of justification for these design choices is a major flaw.

3. Instability of the Curvature Estimate: The core of the method is the quotient h_{k,t} := \Delta g_{k,t} / \Delta \theta_{k,t}. In a stochastic setting, this is notoriously unstable. Both the numerator (gradient change) and denominator (parameter change) are noisy. The paper's primary defense against this is a validity mask m_{k,t} that only checks for |\Delta\theta_{k,t}| > \epsilon. This prevents division by zero but does not protect against a very small, non-zero \Delta\theta_{k,t}, which would cause the quotient to explode. The subsequent clamping \Pi_{[\lambda_{min},\lambda_{max}]}  is a hard, ad hoc fix for this instability, not a principled solution.

4. Lack of Ablation Studies: The algorithm is a collection of clever but unverified engineering tricks. The paper provides no ablation studies to demonstrate that this specific combination is necessary or optimal.

4.1 t-weighting (Eq 16 and Eq 19): The weighted average gives priority to later steps10. Why is this linear t-weighting superior to a simple, unweighted average or a more standard exponential moving average?

4.2 \gamma_{k,t} Annealing (Eq 12): The curvature-aware divisor \gamma_k is linearly annealed back to 1.0 over the course of the next epoch. The justification that the estimate "loses fidelity" is intuitive but hand-wavy. What happens if this annealing is removed and \gamma_{k,t} = \gamma_k is held constant? What if the annealing is exponential instead of linear? This is a core component of the method and it is completely unevaluated.

4.3 Quantile Choice (Q_{\omega}): The use of a low-tail quantile (\omega=0.1) to compute \gamma_k is another heuristic to provide robustness. How sensitive is the method to this choice? What if \omega=0.5 (median) or \omega=0.01 were used? This introduces a critical new hyperparameter without analysis.

5. New Hyperparameters: The method is presented as a "booster," but it introduces at least four new, sensitive hyperparameters: \eta_2 (secondary learning rate), \lambda_{min}, \lambda_{max} (clamping bounds), and \omega (quantile percentile). The authors themselves admit that in cases where CT-AGD is slower, the solution is to tune these parameters, which undermines the "plug-and-play" implication.

**Questions:**

see my comments above

---

> ### Author Response · Authors · 2025-11-20
>
> We thank the reviewer for the constructive feedback and for acknowledging the empirical strengths of CT-AGD.
>
> **Q1**: As was mentioned in the response to reviewer a4gh, we stress that our theoretical analysis was meant solely to show that the existing guarantees were maintained for convex settings. No additional claims were made or meant for general problems. We have revised the manuscript to insure no such claims exist. There is additional overhead compaired to Adam but it is within the same order of magnitude, which is why we termed it minimal when compared to second order methods such as L-BFGS.
>
> **Q2** and **Q3**: Since we are in essence estimating the curvature with a finite-difference approximation, we are expetedly sensitive to sharp changes. The clamping that we implemented is in essence equivalent to a step size shrinking, common in such settings. We have observed that in general epoch averaging of the curvature estimate together with the validity mask already suffices to obtain a stable diagonal signal; clamping in $[\lambda_{\min},\lambda_{\max}]$ primarily enforces positivity and guards against rare outliers. We added ablation information to the manuscript (Section 4.1) that corroborates these findings.
>
> **Q4**: To address the concern that the method is a collection of unverified heuristics, we added targeted ablations in the new subsection 4.1 Ablation Studies of the manuscript that directly probe the annealing schedule and curvature averaging scheme:
>
> **annealing, curvature averaging and robustness on CIFAR-10 / ResNet**
>
> | Type of run | Description | Top-1 % | Epochs to conv. |
> |---|---|---:|---:|
> | No annealing | $\mu_{k,t}=\eta_1/\gamma_{k,t}$ is constant within the epoch. | 50.96 ± 3.82 | 2.20 ± 0.56 |
> | Exponential annealing | Exponential annealing with $\alpha=\tfrac{1}{2}$. | 90.32 ± 0.35 | 27.40 ± 5.31 |
> | Baseline | Linear annealing + weighted curvature estimation. | 90.01 ± 0.22 | 26.20 ± 4.16 |
> | Non-weighted curvature | Curvature estimate based on simple average. | 90.05 ± 0.29 | 26.80 ± 1.62 |
> |SGD||90.35 ± 0.14 | 44.00 ± 3.29
>
> Removing annealing (“No annealing”) severely degrades accuracy and increases variance, showing that some annealing of the inter-epoch curvature cue is necessary. Both exponential and linear annealing recover high accuracy and similar epochs to convergence, and a simple non-weighted average performs almost identically to the weighted scheme. This indicates that CT-AGD depends on the presence of annealing but not on a finely tuned annealing form, and that the curvature averaging does not need to be carefully engineered.
>
> We also sweep the low-tail quantile $Q_\omega$ used to define $\gamma_k$ and observe that CT-AGD remains stable across a broad range of values:
>
> | $\omega$ | Top-1 % | Epochs to conv. |
> |---:|---:|---:|
> | 0.1 | 90.23 ± 0.22 | 25.80 ± 1.25 |
> | 0.2 | 90.16 ± 0.26 | 23.80 ± 3.21 |
> | 0.5 | 90.18 ± 0.11 | 35.80 ± 3.87 |
> |SGD|90.35 ± 0.14 | 44.00 ± 3.29
>
> These experiments show that $\omega$ does not critically affect final accuracy and mainly modulates how aggressively CT-AGD exploits the curvature cue when trading off convergence speed. Together with the interval/clamping sweeps shown in the response to reviewer pC3E, they support our claim that CT-AGD is robust to its main heuristic choices and behaves as a practical booster that consistently reduces epochs to convergence without heavy hyperparameter tuning.
>
> **Q5**: We agree with the reviewer that introducing new hyperparameters without presenting ablation studies can raise realistic concerns about the applicability of the method without considerable fine-tunning. For this reason, we have updated the manuscript to include the ablations previously mentioned which provide some level of confidence in the robustness of the method, reducing the concern that thorough fine-tunning is required to achieve the results presented.

---

### Official Review · Reviewer_a4gh · 2025-10-29

**Soundness:** 1
**Presentation:** 2
**Contribution:** 2
**Rating:** 2
**Confidence:** 4

**Summary:**

This paper is concerned with accelerated gradient descent. The core idea of the paper is to make use of some estimation of the local curvature of the function to minimize to accelerate the speed of convergence.

**Strengths:**

* The considered problem is a main bottleneck in deep learning. How to accelerate the learning phase ? First order stochastic methods such as SGD  (a famous variation beeing ADAM algorithm) have proven to be the most interesting type of methods. However, there is still place for improvements in term of speed, theoretical guarantees, ...

* Making use of the local curvature to accelerate first order methods is a sound idea. This is for instance the main ingredient in the famous L-BFGS algorithm. This can be seen as an automated time step algorithm.

* Estimation of the local curvature can be computationally costly. This is the main reasons why heuristics are introduced in this paper to get a good approximation without too much computations.

**Weaknesses:**

* The paper mainly relies on a heuristic to estimate the local curvature. However, the theoretical results of the paper are weak the proof of Theorem 2.1 seems to be a direct adaptation of one of the proofs of the review paper, once the estimation of the Hessian has been clipped, see
Handbook of Convergence Theorems for (Stochastic) Gradient Methods by Garrigos and Grower
https://arxiv.org/pdf/2301.11235

* The paper claims to solve all the deep learning problems. However, the setting of Theorem 2.1 is the convex setting, which of course is an assumption that is not satisfied in practice.

* The writing of the paper is quite poor. Notations change from one section to another. $\bar{x_N}$ in Theorem 2.1 does not seem to be even defined in the main part of the paper.

* The authors have acknowledged having used LLMs for bibliography reasearch. This is probably the reason why they cite the archiv preprint by Garrigos and Grower
https://arxiv.org/pdf/2301.11235
as beeing authored by Garrigos, Gidel, and Dieuleveut.
The archiv preprint number they give in reference is indeed the one by Garrigos and Grower. I could not find any archiv preprint by Garrigos, Gidel and Dieuleveut, so this is probably a nice example of hallucination of their LLM.
I think the authors should be very cautious when using LLMs for bibliography research, and at least double check the LLMs results.

**Questions:**

* By clipping the estimation of the Hessian, you ensure the existence of $\gamma_\min$ and $\gamma_\max$ at each iteration.
However, how do you ensure that the bounds are uniform through the iterations (i.e. that they do not depend on $k$) ?

* There is an inner loop in the algorithm (to estimate the local curvature). The proof of convergence assumes that no error is mase in this inner loop. How robust is the algorithm to errors within this inner loop ?

* What is the intuitive meaning of equation (26) in Theorem 2.1 ? What is the behavior of the sums of $a_l$ and $b_l$ ? Is this a fast convergence ?
The convergence rate is an ergodic convergence rate. What can be said about $x_N$ ?

* There are exeperimental results on 3 data-sets. Since the claim of the paper is that it accelerates over ADAM, I think that there should be more experiments.

---

> ### Author Response · Authors · 2025-11-20
>
> We thank the reviewer for the careful reading, for emphasizing both the importance of curvature information in first order methods and the points where our theory, writing and experiments can be improved.
>
> **Q1**: we will remove $\gamma_{min}$ and $\gamma_{max}$ mentions and switch them to references to $\lambda_{min}$ and $\lambda_{max}$, which should make it clear that the bounds are uniform through the iterations.
>
> **Q2**: we did additional ablation studies where we add an unbiased noise term to each quotient to ensure the errors don't accumulate. The results are presented in the new subsection 4.1 Ablation Studies of the manuscript and show that the method's averaging and clamping makes it rubust to errors.
>
> |  Noise $\sigma^2$ | Top-1 (%) | Epochs to conv. |
> |---:|---:|---:|
> | 0 | $90.09 \pm 0.16$ | $23.67 \pm 1.49$ |
> | $0.01$ | $90.07 \pm 0.23$ | $25.80 \pm 2.39$ |
> | $0.1$ | $90.12 \pm 0.14$ | $24.80 \pm 1.04$ |
> |SGD|$90.35 \pm 0.14$ | $44.00 \pm 3.29$
>
> **Q3**: the coefficients $a_\ell$ are the “descent weights” proportional to the effective step sizes and encode how much each iterate contributes to progress, while the $b_\ell$ collect the accumulated variance terms. For typical choices of steps, for instance $\mu_\ell \propto 1/\sqrt{\ell}$ with matched boundary steps, one obtains $\sum a_\ell \asymp \sqrt{N}$ and $\sum b_\ell \asymp \log N$, which yields the standard stochastic convex rate $\mathbb E[f(\bar x_N)-f^\star] = O\big(\log N / \sqrt{N}\big)$. This is not faster in order than the optimal SGD, the faster convergence is not derived theoretically but showed in the experimental and tool results. Our current statement focuses on the weighted average because it arises directly from the telescoping argument and is standard in stochastic convex analysis.
>
>
> **Q4**: we expanded the experiments with three additional lightweight setups: a GraphSAGE GNN and a feature-only MLP on the Coauthor CS benchmark, and a bidirectional LSTM sentiment classifier on IMDB. An additional larger experiment is being conducted and once its results are available we will update the manuscript.The current results further illustrate the effectiveness of our method (see Section S.6 Aditional Experiments in the manuscript) and the tables below:
>
> **Coauthor CS – GraphSAGE**
>
> | Method          | Top-1 (%)           | Epochs to conv.      |
> |:----------------|--------------------:|---------------------:|
> | Adam            | $93.90 \pm 0.20$    | $1.00 \pm 0.00$      |
> | CT-AGD (Adam)   | $94.00 \pm 0.30$    | $1.00 \pm 0.00$      |
> | CT-AGD (SGD)    | $86.80 \pm 0.30$    | $4.30 \pm 0.50$      |
> | SGD             | $86.50 \pm 0.40$    | $5.20 \pm 0.40$      |
>
> **Coauthor CS – MLP**
>
> | Method          | Top-1 (%)           | Epochs to conv.      |
> |:----------------|--------------------:|---------------------:|
> | Adam            | $92.80 \pm 0.30$    | $2.70 \pm 0.50$      |
> | CT-AGD (Adam)   | $92.80 \pm 0.50$    | $2.50 \pm 0.50$      |
> | CT-AGD (SGD)    | $88.50 \pm 1.50$    | $6.00 \pm 0.90$      |
> | SGD             | $87.30 \pm 2.60$    | $6.30 \pm 1.50$      |
>
> **IMDB – LSTM sentiment classifier**
>
> | Method          | Top-1 (%)           | Epochs to conv.      |
> |:----------------|--------------------:|---------------------:|
> | Adam            | $86.20 \pm 0.40$    | $1.40 \pm 0.50$      |
> | CT-AGD (Adam)   | $86.30 \pm 0.40$    | $1.40 \pm 0.50$      |
> | CT-AGD (SGD)    | $87.30 \pm 0.30$    | $2.20 \pm 0.40$      |
> | SGD             | $86.80 \pm 1.20$    | $2.00 \pm 0.00$      |
>
>
> Regarding the theoretical novelty and the remark that Theorem 2.1 is a direct adaptation of Garrigos and Gower, our analysis differs in the step-dependent diagonal matrix $P_k$ and the curvature cue $\gamma$. Still, since most steps are adaptive SGD, the proof is ultimately an extension of their established convergence result and our goal was simply to show that the existing guarantees are still maintained. Concerning the scope of Theorem 2.1 and the impression that we “solve all deep learning problems,” we stress that our main theorem assumes convexity and is not in any way a general guarantee for arbitrary deep models. We will harmonize notation throughout and correct the citation. We thank the reviewer for noting this obvious error.

---

### Official Review · Reviewer_pC3E · 2025-10-31

**Soundness:** 3
**Presentation:** 3
**Contribution:** 2
**Rating:** 4
**Confidence:** 4

**Summary:**

The paper develops what is called a Curvature-Tuned Accelerated Gradient Descent technique for deep learning, CT-AGD for short. The heuristic technique is a cheap and clever approach to capture the direction of local curvature via first-order differencing and the use of other clever ideas. The authors claim this approach mitigates noise and bias introduced by stochastic mini-batch training. They also claim the heuristic technique introduces minimal storage and computational overhead to the underlying stochastic gradient algorithms. Experiments on Vision tasks were used to show effectiveness at similar level of accuracy.

**Strengths:**

I think the strength of the paper which stands out to me, is the cleverness of the technique, especially in the way they combine different ideas that form the technique.

**Weaknesses:**

1. The paper, like lot of papers follow the trend of interpreting their techniques as preconditioners, when they really are not.
They justify this by using the term 'first-order methods' for the stochastic gradient algorithms under consideration.
However, closely looking at these techniques, including the current paper, they can more simply and better viewed as a choice of iterative learning rates, without overclaiming.
In the case of this paper, I think the preconditioning interpretation is one of its weaknesses. The preconditioning framing of the paper while recently popular can be extremely misleading and often obscures the simplicity of the methods, and detracts from intuitive understanding

2. Another weakness lies in the treatment of the interval thresholds $\lambda_{\min}$ and $\lambda_{\max}$ which are passed off as bounds of the diagonal entries of the true Hessian. These thresholds are not estimated, even if unknown. Instead, they are chosen as hyperparameters which introduces a degree of arbitrariness and opens up a wide range of possibilities that limits the robustness and generalizability of the technique in relation to the underlying methods when compared with another well-tuned learning rate (iterative or fixed).

3. The last weakness is the claim that the proposed technique mitigates noise and bias introduced by stochastic mini-batch training. But these is more attributed to the role of gradient smoothing, which has been called momentum.  Even without momentum, the stochastic gradient algorithm is competitive in these kinds of CNN architecture experiments. In addition, the extent to which the technique boosts the underlying method is marginal.

**Questions:**

Nil

---

> ### Author Response · Authors · 2025-11-20
>
> We thank the reviewer for their careful reading of our work and for highlighting both the strengths and the potential weaknesses of the current presentation.
>
> **Q1**: Regarding the preconditioning interpretation, we agree that our method can be seen as a coordinate wise, curvature informed learning rate scheme and we will revise the manuscript accordingly.
>
> **Q2**: Concerning the interval thresholds $(\lambda_{\min}, \lambda_{\max})$, we acknowledge that our original phrasing may wrongly suggest that these quantities act as bounds on the true diagonal of the Hessian. Our intention is to ensure we have clamps on a possibly noisy diagonal curvature proxy $\widehat H$, to enforce positivity and limit rare outliers when forming $P = 1 \oslash \widehat H$. To study the impact of the arbitrarily chosen $(\lambda_{\min}, \lambda_{\max})$  we reproduce below the corresponding ablation (new subsection 4.1 Ablation Studies of the manuscript) showing that performance is stable when varying these thresholds over several orders of magnitude (CIFAR-10, ResNet-18):
>
> | $\lambda_{\min}$ | $\lambda_{\max}$ | Top-1 (%) | Epochs to conv. |
> |---:|---:|---:|---:|
> | $10^{-1}$ | $10^{1}$ | $90.13 \pm 0.15$ | $24.60 \pm 2.86$ |
> | $10^{-2}$ | $10^{2}$ | $90.01 \pm 0.22$ | $26.20 \pm 4.16$ |
> | $10^{-3}$ | $10^{3}$ | $90.23 \pm 0.22$ | $25.80 \pm 1.25$ |
> |SGD||$90.35 \pm 0.14$ | $44.00 \pm 3.29$|
>
> This experiment illustrates that the method’s performance and convergence speed vary only mildly across a wide range of $(\lambda_{\min}, \lambda_{\max})$, which directly addresses the reviewer’s concern about robustness and arbitrariness and shows that these thresholds do not require fine tuning beyond the usual care.
>
> **Q3**: Finally, regarding the claim about mitigating noise and bias in stochastic mini batch training, we agree that our original wording could be read too broadly. We do not intend to suggest that our method produces less noisy updates than other first order methods, the “less noisy” aspect we refer to is specifically relative to second order approaches that estimate curvature from one or a few mini batches.

---

> ### Comment · Reviewer_pC3E · 2025-11-25
> **Response to Authors**
>
> I understand that Q1 and Q3 would be resolved by revising the paper where necessary.
>
> For Q2. I understand that the intent was to clamp the estimate of concern here.
> However, one interpretation given the ablations, is that $\lambda_{max}$ is likely unnecessary. As you stated, the intention here is just to project the estimates to be non-negative and finite, so as to safeguard the updates from blowing up. However, you need to motivate what you mean by rare outliers in the case of the estimates without knowing the true bounds.
>
> Some thoughts, have you tried using [$\lambda_{min}$, $\lambda_
> {max}$] = [1e-5, 1e-1] or [1e2, 1e5].

---

> > ### Author Response · Authors · 2025-12-02
> >
> > We ran additional experiments on CIFAR-10 with ResNet-18 using our CT-AGD optimizer, varying only the curvature clamps:
> >
> > | $\lambda_{\min}$ | $\lambda_{\max}$ | Top-1 (%)           | Epochs to conv.       |
> > |-------------------:|-------------------:|---------------------|------------------------|
> > | $10^{-2}$        | none               | $90.37 \pm 0.39$  | $25.75 \pm 3.01$     |
> > | $10^{2}$         | $10^{5}$         | $85.06 \pm 0.29$  | $99.00 \pm 6.63$     |
> >
> > We also tested the suggested $[\lambda_{min}, \lambda_{max}] =$ [1e-5, 1e-1], and in those cases CT-AGD diverged after the first boundary step.
> >
> > These results show that:
> > 1. Removing the upper clamp $\lambda_{\max}$ does not cause instabilities and preserves good performance and convergence.
> > 2. Overly aggressive clamping (large $\lambda_{\min}$, large $\lambda_{\max}$) significantly harms both accuracy and speed because the effective LR is forced to be too small.
> > 3. Very small clamps (both $\lambda_{\min}$ and $\lambda_{\max}$ close to zero) make the boundary step explode (step size $\propto 1/\lambda$), which in practice causes CT-AGD to diverge after the first boundary step.
> >
> > In practice, we only need a small $\lambda_{\min}$ to avoid division by tiny curvature estimates, while an explicit $\lambda_{\max}$ is unnecessary. We thank the reviewer for this insightful suggestion, which helped us both simplify and clarify the approach.

---

### Author Response · Authors · 2025-12-02
**Additional Experiment**

We have added a new experiment on large-scale node classification using a 2-layer GraphSAGE model on the Reddit benchmark (Supplemental Section S.6). As summarized in the table below, CT-AGD (Adam) matches the strong performance of Adam with identical convergence behavior, while CT-AGD (SGD) improves both training accuracy and convergence speed relative to standard SGD. These results further support the conclusion that CT-AGD consistently shows benefits over its first-order backbones across architectures and data regimes.

**Table S.6: Reddit / GraphSAGE Results**

Method                         | Train Acc. (%)     | Epochs to Conv.
|---:|---:|---:|
Adam                | 97.60 ± 0.00        | 23.4 ± 0.5
CT-AGD (Adam)        | 97.60 ± 0.00        | 23.2 ± 0.4
CT-AGD (SGD)          | 93.30 ± 0.00        | 18.8 ± 0.8
SGD                 | 92.80 ± 0.00        | 27.8 ± 0.4

---

### Meta-Review · Area_Chair_wgeL · 2026-01-11

**Summary:**

Across reviews, the main positive assessment is that the proposed CT-AGD mechanism is technically clever and shows empirical speedups (fewer epochs to reach target accuracy) while maintaining comparable final accuracy on the reported benchmarks. However, the suggested decision is reject because several concerns remain decision-critical. First, the theoretical analysis is largely disconnected from the paper’s intended use case: it provides convex guarantees that do not explain or substantiate the claimed benefits in non-convex deep learning training, and the rebuttal confirms that the theory is mainly intended to preserve existing convex guarantees rather than establish acceleration. Second, although the rebuttal adds meaningful ablations to justify several heuristic design choices, the resulting picture is that the method’s performance depends on key heuristics (notably annealing), and the overall contribution remains more engineering-driven than theoretically grounded. Third, there are unresolved concerns about how well the work is contextualized with respect to recent optimizer literature and modern baselines, which affects confidence in the novelty and in the strength of the empirical claims for an ICLR 2026 accepted submission.

**Reviewer Concerns:**

### Concerns substantially addressed by the rebuttal:
The rebuttal meaningfully improved the empirical support around the method’s heuristic components. In particular, it added targeted ablations for (i) annealing (showing that removing annealing degrades accuracy and increases variance, while different annealing forms perform similarly), (ii) curvature averaging (weighted vs simple averaging behaving similarly), (iii) quantile choice for the curvature cue (showing stability across a range with a speed–aggressiveness trade-off), and (iv) curvature clamping (including evidence that an upper clamp may be unnecessary while a small lower clamp helps avoid instability). These additions directly address the core requests from FzcT and pC3E for justification and sensitivity analysis, and they partially mitigate the concern that the method is an unverified collection of tricks. The rebuttal also commits to revise framing that could be read as overclaiming (e.g., interpreting the method as a preconditioner in a misleading way, or suggesting broadly reduced mini-batch noise), which likely resolves pC3E’s and partially resolves related presentation concerns.


### Concerns only partially addressed or still outstanding:
- The theory–practice mismatch remains largely unresolved. Reviewers a4gh, FzcT, and DdLg raised that convex analysis does not match the non-convex deep learning setting and provides limited insight into why acceleration occurs. The rebuttal mainly narrows claims (theory is for convex settings to show guarantees are maintained) rather than providing a non-convex analysis or a stronger mechanism-based explanation; therefore, the theoretical contribution remains limited and does not support the core empirical motivation.

- In addition, concerns about literature coverage and contemporary contextualization remain: DdLg highlighted that the related work and comparisons appear outdated for ICLR 2026, and the rebuttal does not convincingly demonstrate a substantial update to recent optimizer advances or a clearer positioning relative to modern baselines.

- Finally, questions about scalability and practical overhead are only partially mitigated. While the rebuttal argues overhead is comparable in order to Adam and far less than second-order methods, the work does not yet convincingly establish practicality in larger-scale training regimes or show that the method’s benefits persist under modern large-model settings, which was a central part of the

**Reviewer Scores:**

Reviewer pC3E (4: marginally below acceptance): Likely to increase modestly after discussion. The rebuttal clarifies the framing and provides additional evidence that the clamping can be simplified (upper clamp unnecessary) while maintaining stability. A plausible score change is 4 to 6.

Reviewer a4gh (2: reject): Likely to improve slightly but remain a reject. The rebuttal fixes the citation issue in principle, explains the bounds as uniform clamps, adds robustness ablations, and expands experiments. However, the reviewer’s central objections about weak/derivative convex theory and mismatch to deep learning remain.

Reviewer FzcT (4: marginally below acceptance): Likely to increase. The rebuttal directly addresses the main critique (lack of ablations and justification for heuristic design) with targeted experiments and sensitivity analysis. Remaining concerns about theory relevance may persist but were not the dominant factor in this review. A plausible change is 4 to 6.

Reviewer DdLg (0: strong reject): Unlikely to remain, though it could become slightly less extreme. The rebuttal does not convincingly address the major concerns about outdated related work, modern baseline expectations, and large-scale practicality.

---

### Decision · Program_Chairs · 2026-01-26

Reject